# Seeing, Listening, Remembering, and Reasoning: A Multimodal Agent with Long-Term Memory

**Lin Long**[2,†,*], **Yichen He**[1,*], **Wentao Ye**[2,†], **Yiyuan Pan**[3,†], **Yuan Lin**[1,‡], **Hang Li**[1], **Junbo Zhao**[2], **Wei Li**[1]

[1] Bytedance Seed
  {hyc,linyuan.0,lihang.lh,liwei.85}@bytedance.com

[2] Zhejiang University
  {llong,yewt01,j.zhao}@zju.edu.cn

[3] Robotics Institute, Carnegie Mellon University
  yiyuanp@andrew.cmu.edu

## Abstract

We introduce M3-Agent, a novel **m**ulti**m**odal agent framework equipped with long-term **m**emory. Like humans, M3-Agent can process real-time visual and auditory inputs to build and update episodic and semantic memories, gradually accumulating world knowledge. Its memory is organized in an entity-centric, multimodal manner, enabling deeper and more consistent understanding of the environment. Given an instruction, M3-Agent autonomously performs multi-turn reasoning and retrieves relevant memories to complete tasks. To evaluate memory effectiveness and memory-based reasoning in multimodal agents, we develop M3-Bench, a long-video question answering benchmark comprising 100 newly recorded robot-perspective videos (M3-Bench-robot) and 920 diverse web-sourced videos (M3-Bench-web). We annotate QA pairs designed to test capabilities essential for agent applications, such as person understanding, general knowledge extraction, and cross-modal reasoning. Experimental results show that M3-Agent, trained via reinforcement learning, outperforms the strongest baseline, a prompting agent using Gemini-1.5-pro and GPT-4o, achieving 6.7%, 7.7%, and 5.3% higher accuracy on M3-Bench-robot, M3-Bench-web and VideoMME-long, respectively. Our work advances multimodal agents toward more human-like long-term memory and provides insights for their practical design. Models, datasets and code are available at https://github.com/ByteDance-Seed/m3-agent.

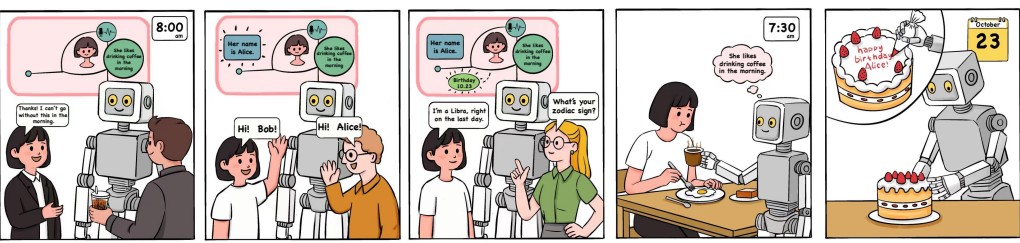

*Multimodal agents continuously perceive their environment, build entity-centric, multimodal long-term memories, and reason over them.*

---

*Equal contribution.

‡Corresponding author.

†Work done during ByteDance Seed internship.

## 1 INTRODUCTION

Imagine that in the future a household robot can autonomously carry out household tasks without your explicit instructions; it must have learned the operational rules of your home through daily experiences. In the morning, it hands you a cup of coffee without asking "coffee or tea?", because it has gradually formed a memory of you, tracking your preferences and routines through long-term interactions. For a multimodal agent, achieving this level of intelligence fundamentally relies on three capabilities: (1) continuously perceiving the world via multimodal sensors; (2) storing and organizing its experiences into a long-term memory, and gradually building knowledge of its environment; (3) reasoning over this accumulated memory to guide its actions.

To achieve the goals, we propose M3-Agent, a novel **m**ulti**m**odal agent framework equipped with long-term **m**emory. As shown in Figure 1, it operates through two parallel processes: **memorization**, which continuously perceives real-time multimodal inputs to construct and update long-term memory; and **control**, which interprets external instructions, reasons over the stored memory, and executes the corresponding tasks.

During memorization, M3-Agent processes the incoming video stream, capturing both fine-grained details and high-level abstractions by generating two types of memory, analogous to human cognitive systems (Tulving, 1972; 1985): **Episodic memory** corresponds to concrete events observed within the video (e.g., *"Alice takes the coffee and says, 'I can't go without this in the morning,'"* and *"Alice throws an empty bottle into the green garbage bin."*), and **semantic memory** refers to general knowledge that can be derived from the clip (e.g., *"Alice prefers to drink coffee in the morning"* and *"The green garbage bin is used for recycling."*).

The generated contents are then integrated into the agent's long-term memory, which supports multimodal information such as faces, voices, and textual knowledge. The memory is organized in an entity-centric structure. For example, information related to the same person (e.g., their face, voice, and associated knowledge) is connected within a graph, as shown in Figure 1. These connections are incrementally established as the agent extracts and integrates episodic and semantic memory.

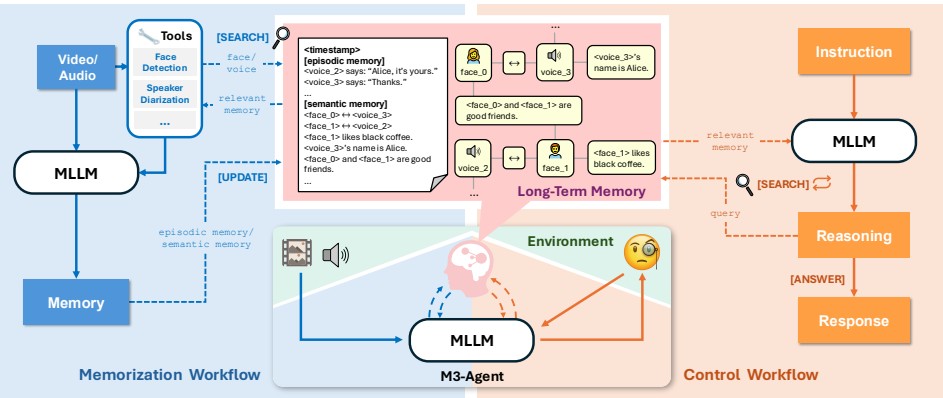

Figure 1: Architecture of M3-Agent, comprising a multimodal large language model (MLLM) and a multimodal long-term memory. It supports two parallel processes: **memorization** and **control**. The entity-centric memory structure enables accumulate world knowledge and maintain consistent, context-rich memory over time.

During control, M3-Agent leverages its long-term memory to reason and complete tasks. It autonomously retrieves relevant information from its memory across different dimensions (e.g., events and characters). Unlike standard retrieval-augmented generation (RAG) (Lewis et al., 2020), which performs single-turn memory injection into the context, we apply RL in M3-Agent to enable multi-turn reasoning and iterative memory access (Jin et al., 2025), resulting in more focused and context-aware retrieval.

Such a memory-augmented system extends beyond conventional long video description (Han et al., 2023; Zhang et al., 2024a; Islam et al., 2024), presenting two key challenges: (1) **Infinite information processing**. Existing methods for long video understanding optimize architectural efficiency to process longer, but still finite, offline videos (Song et al., 2024; Shen et al., 2024; Zhang et al.,

2024b; He et al., 2024a; Shu et al., 2025). In contrast, M3-Agent continuously processes arbitrarily long streams in an online manner, more closely mimicking how human long-term memory forms, through ongoing perception and incremental experience integration. (2) **World knowledge construction**. Traditional video description (Li et al., 2023; Lin et al., 2023a; 2024; Yuan et al., 2025; Bai et al., 2025) often focuses on low-level visual details while overlooking high-level world knowledge (Qiao et al., 2024; Ivanova et al., 2024; Fung et al., 2025) such as character identity and entity attributes, which may lead to ambiguity and inconsistency in long-term contexts. M3-Agent addresses this by incrementally building an entity-centric memory that encodes rich multimodal representations of key entities, enabling coherent and consistent long-term memory.

We evaluate M3-Agent on long video question answering (LVQA), using long videos to simulate the multimodal input streams received by agents. Existing LVQA benchmarks (Fu et al., 2025; Zhou et al., 2025; Chandrasegaran et al., 2024; Wu et al., 2024) mainly target on visual perception (e.g., action recognition, spatial/temporal perception) but overlook higher-level cognitive abilities that rely on long-term memory and are crucial for real-world agents, such as person understanding, general knowledge extraction, and cross-modal reasoning. To fill this gap, we introduce M3-Bench, a new LVQA benchmark designed to evaluate a multimodal agent's ability to reason over memory. M3-Bench includes two subsets: (1) M3-Bench-robot, with 100 real-world videos recorded from a robot's perspective, and (2) M3-Bench-web, with 920 YouTube videos covering diverse scenarios. We define five question types (Table 1) targeting different aspects of memory-based reasoning, and annotate 1,276 and 3,214 QA pairs for M3-Bench-robot and M3-Bench-web, respectively.

We conduct experiments on M3-Bench-robot, M3-Bench-web, and VideoMME-long (Fu et al., 2025). M3-Agent, trained via reinforcement learning, outperforms all baselines on all three benchmarks. Compared to the strongest baseline, Gemini-GPT4o-Hybrid, which implements M3-Agent framework by prompting advanced proprietary models for both memorization and control, M3-Agent achieves 6.7%, 7.7%, and 5.3% higher accuracy on M3-Bench-robot, M3-Bench-web, and VideoMME-long, respectively. Extensive ablation studies further demonstrate the importance of long-term memory, particularly semantic memory, which substantially improves accuracy across all tasks. Ablation studies also highlight the significance of reinforcement learning and multi-turn reasoning for effective control process, enabling M3-Agent to solve more complex QA tasks.

The main contributions of this paper are summarized as follows:

- We introduce M3-Agent, a novel framework for multimodal agents with long-term memory. M3-Agent continuously processes real-time multimodal inputs (**seeing** and **listening**), incrementally builds world knowledge by generating episodic and semantic memories (**remembering**), and reasons over these memories to complete complex question answering tasks (**reasoning**).
- We develop M3-Bench, a new LVQA benchmark designed to evaluate the effectiveness of memory and memory-based reasoning for multimodal agents.
- Our experiments demonstrate that M3-Agent, trained by reinforcement learning, consistently outperforms other baselines across multiple tasks.

## 2 RELATED WORK

**Long-Term Memory of AI Agents.** Long-term memory is essential for AI agents (Feng et al., 2024). A common approach to constructing memory is to append complete agent trajectories, such as dialogues (Mei et al., 2024; Wang et al., 2023; Liu et al., 2024a; Zhong et al., 2024) or execution trajectories (Liu et al., 2024c; Wang et al., 2024b; Shang et al., 2024; Liu et al., 2024a; Hu et al., 2024; Sarch et al., 2023), sometimes enhanced with summarization (Wang et al., 2023; Li et al., 2024a; Hu et al., 2024; Zhong et al., 2024), latent embeddings (Zhang et al., 2024b; Liu et al., 2024b; Song et al., 2024; Diko et al., 2025), or structured knowledge representations (Ocker et al., 2025; Xu et al., 2025b). Chhikara et al. (2025); Wang et al. (2023); Kang et al. (2025) introduce more sophisticated memory architectures for fine-grained control. While most existing efforts focus on LLM agents, multimodal agents process richer, more diverse inputs, which poses challenges in maintaining consistent long-term memory Fan et al. (2024b;a). Moreover, their ability to construct world knowledge through lived experience, rather than static description, remains unexplored.

**Online Video Understanding.** For multimodal agents, memory formation is closely related to online video understanding, which requires real-time stream processing and decision-making based on past observations. Traditional methods, such as extending context windows (Chen et al., 2024b;

Zhang et al., 2024d) or compressing visual tokens (Wu et al., 2022; Lan et al., 2024; Wu et al., 2022), do not scale to infinite streams, as reprocessing full histories for each instruction is computationally prohibitive. Memory-based methods (Zhang et al., 2024b; He et al., 2024a; Song et al., 2024; Zhang et al., 2024c) improve scalability by storing encoded visual features for retrieval; but they often fail to ensure long-term consistency, losing track of identities or evolving events. With the rise of large multimodal and language models (Hurst et al., 2024; Team et al., 2024; Yang et al., 2025; Bai et al., 2025; Yuan et al., 2025), the Socratic Models framework (Zeng et al., 2022; Lin et al., 2023b; Zhang et al., 2024a) offers a scalable alternative by generating language-based video memories. VideoAgent (Fan et al., 2024b) employs prompt engineering to enable agentic workflows for long video understanding. Despite this, it still struggles to maintain consistency over complex, evolving content and lacks a clear training and optimization approach. StoryTeller (He et al., 2024b) introduces a method for person identity tracking in long videos, but it requires full access to the entire video, making it unsuitable for streaming scenarios.

## 3 DATASETS

We first introduce M3-Bench, an LVQA dataset designed to evaluate the capability of multimodal agents to perform reasoning over long-term memory. Each instance in M3-Bench comprises a long video simulating the perceptual input of an agent, along with a series of open-ended question-answer pairs. The dataset is organized into two subsets: (1) M3-Bench-robot, which contains 100 real-world videos recorded from a robot's egocentric perspective, and (2) M3-Bench-web, which includes 920 web-sourced videos covering a wider variety of content and scenarios. To comprehensively assess an agent's ability to recall past observations and perform memory-based reasoning, we curate five distinct types of questions, as summarized in Table 1. Overall, M3-Bench is featured by (1) long-duration, real-world videos that encompass diverse real-life scenarios relevant to the deployment of multimodal agents, and (2) challenging questions that extend beyond shallow perceptual understanding and require complex reasoning over long-term contexts.

Table 1: Explanations of different question types and their corresponding examples in M3-Bench.

| Question Type | Explanation and Example |
|---|---|
| Multi-evidence Reasoning | This requires aggregating multiple pieces of information distributed across the video. **Example:** *Which collection has the highest starting price among the five items shown in the video?* The agent must identify the starting price from five segments and compare them to find the highest. |
| Multi-hop Reasoning | This involves step-by-step reasoning across different segments to reach a conclusion. **Example:** *Which bubble tea shop did they visit after going to Ding Cha?* The agent must first locate the visit to Ding Cha, then follow subsequent segments to identify the next bubble tea shop. |
| Cross-modal Reasoning | This requires reasoning across multiple modalities, such as visual and audio content. **Example:** *(Bob shows Robot a red folder and says, "Confidential documents go in this folder," then shows a white folder and says, "Normal documents go in this one.") Which folder should confidential documents be placed in?* The agent must combine visual cues (folder color) and verbal cues to answer. |
| Person Understanding | This involves reasoning about person-related attributes such as identity, emotions, personality, or relationships. **Example:** *Is Lucas skilled at cooking?* The video does not directly reveal the answer, but the agent must aggregate Lucas's behavior across multiple cooking scenes to infer his skill level. |
| General Knowledge Extraction | This evaluates whether the agent can extract general knowledge from specific events. **Example:** *(A person has classified different groceries into various shelves of a refrigerator) Which shelf is suitable for storing vegetables?* The agent must recognize typical storage rules from its observation to answer the question. |

Figure 2 shows examples from M3-Bench. Figure 3 presents overall dataset statistics, and Table 2 compares M3-Bench with existing LVQA benchmarks. The rest of this section details the data collection and annotation processes for M3-Bench-robot and M3-Bench-web.

### 3.1 M3-BENCH-ROBOT

We first focus on robots, one of the most representative embodiments of multimodal agents. A general-purpose robot should maintain long-term memory and reason with it to guide its actions. For example, a household robot might need to remember a person's name or their coffee preferences. Reasoning over such memory supports higher-level cognition, such as inferring personality traits, social relationships, and object functions. To evaluate these abilities, we construct a complete data pipeline—from recording egocentric videos from a robot's perspective to manual QA annotation.

**Script Design** We first design video scripts across seven everyday settings: living room, kitchen, bedroom, study, office, meeting room, and gym. Each script involves one robot interacting with

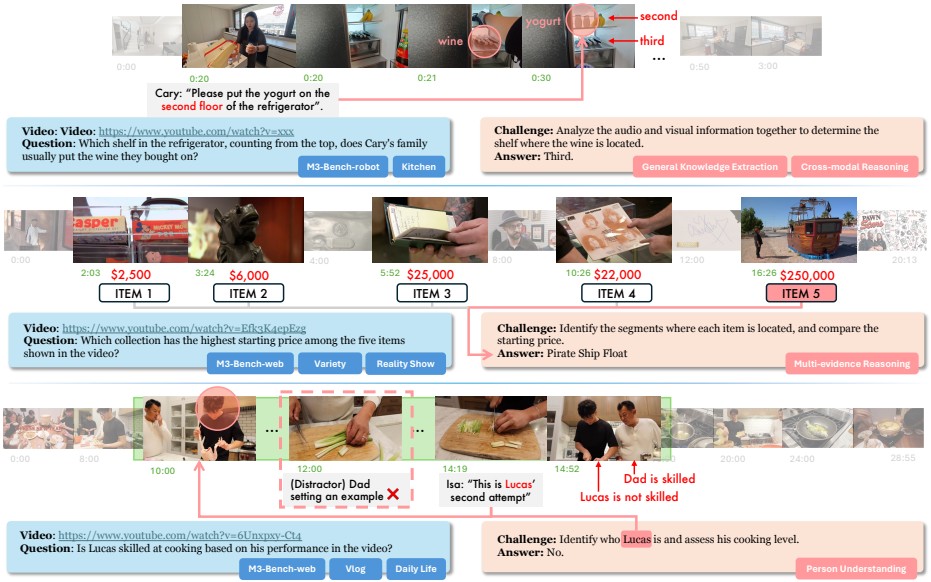

Figure 2: Examples from M3-Bench.

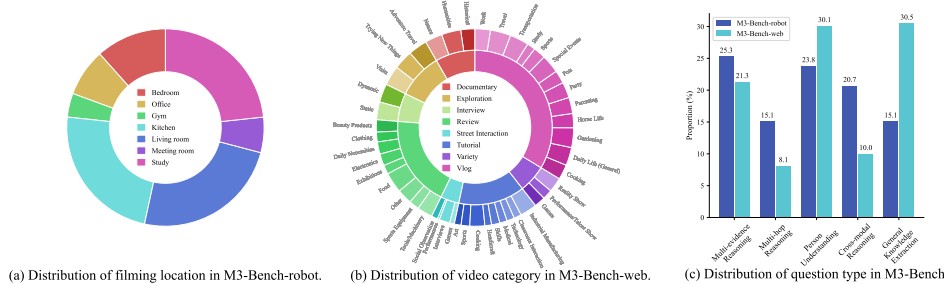

(a) Distribution of filming location in M3-Bench-robot. (b) Distribution of video category in M3-Bench-web. (c) Distribution of question type in M3-Bench.

Figure 3: M3-Bench Statistics. Each question may correspond to multiple question types.

two to four humans. To promote diversity, we create multiple thematic variations for each scenario (e.g. family talks or holiday parties in the living room). Each script consists of at least 70 sequential events with at least 15 questions belong to types in Table 1. An example is shown in Table 7 (§ B.6).

**Video Filming** Recording with real robots is costly and complex. Therefore, we employ human actors to simulate robot behavior. For each script, one actor acts as the robot to perform daily tasks, while wearing a head-mounted camera to capture egocentric video and audio. To ensure diversity, we recruit 67 actors and film across 51 locations, with no more than three videos recorded per site.

**Annotations** After recording the videos, annotators curate QA pairs by reviewing the pre-scripted questions. Since some questions become invalid due to deviations during filming, annotators determine whether to retain, revise, or discard each one accordingly. For valid questions, they also annotate the timestamp at which the question should be asked. Additional questions (conforming to types in Table 1) are created when necessary to ensure each video includes at least 12 QA pairs.

## 3.2 M3-BENCH-WEB

To further increase video diversity, we collect extra videos from YouTube following existing practice (Fu et al., 2025; Fang et al., 2024; Niu et al., 2025).

**Video Collection** We adopt a question-driven approach for video collection: annotators select videos that could support at least five questions belonging to the types in Table 1, which naturally encourage the inclusion of videos with rich narratives and complex inter-entity dynamics. To promote video diversity, annotators refer to a predefined list of high-information-density categories relevant to real-world multimodal agent applications. Each category allows up to 20 video submissions. Meanwhile,

Table 2: Comparison of M3-Bench with existing LVQA benchmarks across key aspects: number of videos (**#Videos**), average length in seconds (**Len.**), number of QA pairs (**#QAs**), annotation method (**Anno.**, M/A=manually/automatic), question format (**Form.**, O/C=open-/close-ended), presence of an agent in the video (**Agent Present**), inclusion of questions about cross-modal reasoning (**Cross-Modal QA**), person understanding (**Person QA**), and general knowledge (**Knowledge QA**).

| Benchmark | #Videos | Len.(s) | #QAs | Anno. | Form. | Agent Present | Cross-Modal QA | Person QA | Knowledge QA |
|---|---|---|---|---|---|---|---|---|---|
| EgoSchema (Mangalam et al., 2023) | 5,063 | 180.0 | 5,063 | M/A | C | ✗ | ✗ | ✗ | ✗ |
| LongVideoBench (Wu et al., 2024) | 3,763 | 473.0 | 6,678 | M | C | ✗ | ✗ | ✗ | ✗ |
| HourVideo (Chandrasegaran et al., 2024) | 500 | 2,742.0 | 12,976 | M/A | C | ✗ | ✗ | ✗ | ✗ |
| MVBench (Li et al., 2024b) | 3,641 | 16.0 | 4,000 | A | C | ✗ | ✗ | ✗ | ✗ |
| Video-MME (Fu et al., 2025) | 900 | 1,017.9 | 2,700 | M | C | ✗ | ✗ | ✗ | ✗ |
| MLVU (Zhou et al., 2025) | 1,730 | 930.0 | 3,102 | M/A | O/C | ✗ | ✗ | ✗ | ✗ |
| M3-Bench-robot | 100 | 2,039.9 | 1,276 | M | O | ✓ | ✓ | ✓ | ✓ |
| M3-Bench-web | 920 | 1,630.7 | 3,214 | M | O | ✗ | ✓ | ✓ | ✓ |

annotators are encouraged to propose new categories, provided they are sufficiently distinct from the existing ones. The final dataset comprises 46 distinct video types, as summarized in Figure 3.

**QA Annotations** The same annotator who collect the video also generate at least 5 corresponding QA pairs. Each question must correspond to at least one type in Table 1. Questions are required to be specific, objective, and have a single unambiguous answer that could be reasonably inferred from the video content, to ensure fair and effective evaluation. Those with ambiguous references, such as "the man" or "in the middle part of the video," are deemed invalid. In M3-Bench-web, the timestamp for each question is uniformly set to the end of the video.

Detailed annotation guidelines, annotator information, and quality control procedures for M3-Bench-robot and M3-Bench-web are provided in Appendix B and Appendix C, respectively.

## 3.3 Automatic Evaluation

We use GPT-4o as an automatic evaluator for M3-Bench by prompting it to assess the correctness of generated answers against their corresponding reference answers. The prompt is shown in Table 23 (§ M.1). To validate this approach, we construct a test set of 100 randomly sampled question-reference-answer triples from our method and various baselines (§ 5.1). Three authors independently assess each generated answer, and GPT-4o's judgments are compared with the majority vote of human annotations, yielding 96% agreement and confirming the effectiveness of the approach.

## 4 Approach

### 4.1 Long-Term Memory

Long-term memory in M3-Agent is implemented as an external database that stores information in a structured, multimodal format (text, image, audio). Specifically, memories are organized as an entity-centric multimodal graph, where each node represents a distinct memory item. Each node includes a unique ID, modality type, raw content, reliability weight, embeddings, and other metadata such as timestamps (see Table 8 (§ D) for details). Nodes are connected by undirected edges that represent several types of relationships between memory items. For example, items sharing the same entity ID are linked to represent the entire entity. This design supports not only sequential retrieval of memories based on timestamps but also associative retrieval based on entities.

The agent constructs its memory by incrementally adding new text, image, or audio nodes. If a memory item generated by the memorization process already exists in long-term memory, the corresponding node or edge is reactivated and its weight is increased; otherwise, a new node or edge is created. To handle potential conflicts introduced during memory construction, M3-Agent adopts a weighted voting mechanism at inference time: frequently activated nodes or edges accumulate higher weights and take precedence over less active, conflicting entries. This design promotes robustness and consistency of the memory graph over time.

**Search Tool** To facilitate memory retrieval, we provide a suite of search tools that enable the agent to retrieve relevant memories based on specific requirements. In particular, we implement two types of search mechanisms operating at different levels of granularity, as summarized in Table 3. Detailed implementation of these retrieval mechanisms is provided in Appendix E.

Table 3: Search functions supported by long-term memory.

| Function | Description |
|---|---|
| search_node | Accepts a query and returns the top-$k$ most relevant nodes. Supports multimodal queries (text, image, or audio) and modality-specific retrieval. |
| search_clip | Return top-$k$ most relevant memory clips for a given query. A memory clip refers to the agent's episodic and semantic memory of a segment (typically 30 seconds) generated during clip-by-clip streaming video processing. |

## 4.2 MEMORIZATION

During memorization (Figure 1), M3-Agent processes video stream in a clip-by-clip manner (typically 30s each), generating two types of memory: episodic memory captures raw visual and auditory content; and semantic memory extracts general knowledge such as character identities, attributes, and relationships. Semantic memory enriches content and provides additional cues for retrieval.

**Consistent Entity Representation** A key challenge in constructing high-quality long-term memory is maintaining consistent representations of core concepts, such as main characters and objects, over time. To avoid using ambiguous language-based descriptions (e.g., "a man with a beard" or "a woman in a red dress"), M3-Agent constructs persistent identity representations in its long-term memory using multimodal information. Equipped with tools like facial recognition and speaker identification (see Appendix E), it extracts faces and voices from clips, linking each to an existing node in its long-term memory using search_node function or assigning to a new node. The resulting identifiers (face_id, voice_id) serve as references to the corresponding characters. By leveraging a globally maintained memory graph, M3-Agent enforces consistent character mapping across clips, enabling a coherent long-term memory.

**Memory Generation** Having the face and voice identities, M3-Agent generates both episodic and semantic memories, referencing each character by their face_id or voice_id stored in long-term memory. This ensures each character is consistently grounded in the memory graph. In particular, during semantic memory generation, M3-Agent performs cross-modal reasoning to link faces and voices belonging to the same character, updating the memory graph to connect their respective nodes. Once linked, these nodes are treated as a single character: during retrieval, they are assigned a shared <character_id>, enabling consistent reasoning about characters across modalities. For output format, M3-Agent generates episodic and semantic memory as a list of text entries as shown in Figure 1. Each entry is stored in the memory graph as a node and linked to its associated identities.

---

**Algorithm 1** Control Process

---

**Require:** Input question $q$, policy model $\pi_\theta$, long-term memory $\mathcal{M}$, maximum number of rounds $H$.
**Ensure:** A complete trajectory $\tau$ generated by the agent.
    $\tau \leftarrow$ [{role: "system", content: Format(system_prompt, $q$)},
        {role: "user", content: instruction_prompt}]        ▷ Initialize the trajectory
    $i \leftarrow 0$
    **while** $i < H$ **do**         ▷ Execute up to $H$ rounds
        $\tau_i \leftarrow \pi_\theta(\cdot \mid \tau)$
        Append {role:"assistant", content: $\tau_i$} to $\tau$
        reasoning, action, argument $\leftarrow$ PARSE($\tau_i$)       ▷ Extract action and argument from $\tau_i$
        **if** action = "[Search]" **then**
            memory $\leftarrow$ SEARCH($\mathcal{M}$, argument)     ▷ Search memory using the content as query
        **else**
            Break       ▷ The trajectory ends when action is "[Answer]"
        **end if**
        $i \leftarrow i + 1$
        Append {role: "user", content: memory + instruction_prompt} to $\tau$   ▷ Append search results and prompt for next round
        **if** $i = H - 1$ **then**
            Append {role: "user", content: memory + last_round_prompt} to $\tau$
        **end if**
    **end while**
    **return** $\tau$

---

## 4.3 CONTROL

When an instruction is received, the control process is triggered. As illustrated in Figure 1, during control, M3-Agent autonomously performs multi-turn reasoning and invokes search functions to retrieve relevant memories. Unlike traditional single-turn RAG, this iterative approach enables more complex planning, making the system more flexible and more capable. Specifically, the control process follows Algorithm 1, with prompts in Table 27 (§ M.3). Here $\pi_\theta$ is the control policy, $q$ is user question, and $\mathcal{D}$ is the long-term memory. At each round, $\pi_\theta$ generates a response consisting

of reasoning, an action, and associated argument. If the action is [Search], the system queries $\mathcal{D}$ with the argument and appends retrieved results to the context for the next round. Depending on the context, it can call different search functions to retrieve memories from multiple perspective (e.g., `search_node` for people or `search_clip` for events). If the action is [Answer], the system returns the content and the process terminates. This loop continues for up to $H$ rounds.

### 4.4 TRAINING

Although conceptually, memorization and control can be handled by a single model, here we train two separate policy models to better specialize their respective functions: memorization relies on strong multimodal understanding, while control requires strong reasoning capabilities. To align with these objectives, we initialize the memorization policy with Qwen2.5-Omni (Xu et al., 2025a), which is capable of processing both visual and audio inputs, and the control policy with Qwen3 (Yang et al., 2025), which is known for its advanced reasoning capabilities. The training data is sourced from our in-house video dataset, for which we have permissions for model training. We collect videos along with corresponding question-answer pairs, adhering to the same annotation standards used for M3-Bench-web. In total, the training dataset comprises 500 long videos, corresponding to 26,943 30-second clips, and 2,736 question-answer pairs.

**Memorization** To improve the model's ability to generate desired memories, we perform imitation learning on Qwen2.5-Omni-7b to obtain `memory-7b-sft`. This process is supervised using a synthetic demonstration dataset including (1) *Episodic memory*: For each 30-second clip, GPT-4o is prompted to generate frame-level cues, while Gemini-1.5-Pro provides event-level summaries. The two outputs are merged to create a richer narrative; (2) *Identity mapping*: Based on extracted faces and voices, we design an algorithm to detect high-confidence **meta-clips** (monologues featuring a consistent face and voice) to build a reliable global face–voice identity mapping, which is propagated across relevant clip segments; and (3) *Other semantic memory*: We use a diverse set of prompts shown in Table 10 (§F) to extract various types of semantic memory. In total, we synthesize 10,952 samples and reserve 200 for validation. More details appear in Appendix F.

Based on the demonstration dataset, we perform supervised fine-tuning for 3 epochs with a learning rate of $1e-5$ and a batch size of 16, using 16 GPUs with 80GB memory each.

**Control** We first construct the environment by generating long-term memory for each training video using `memory-7b-sft`. We then apply DAPO (Yu et al., 2025) to train the policy model $\pi_\theta$, initialized from prompting Qwen3-32b (`control-32b-prompt`). For each QA pair $(q, a)$, $\pi_\theta$ rollouts a group of $G$ trajectories $\tau_{i=1}^G$, using the algorithm shown in Algorithm 1. Note that for any given question, the agent is restricted to searching within the memory generated from its associated video. Then, for each trajectory $\tau_i$, the final submitted answer $y_i$ is extracted and evaluated using the GPT-4o evaluator introduced in Section 3.3. The reward of the $i$-th trajectory is given by:

$$R_i = \begin{cases} 1, & \texttt{gpt4o\_evaluator}(q, a, y_i) = \texttt{True} \\ 0, & \text{otherwise} \end{cases} \tag{1}$$

Note that we compute loss only on LLM-generated tokens. The optimization objective is:

$$\mathcal{J}_{\text{DAPO}}(\theta) = \mathbb{E}_{(q,a)\sim\mathcal{D}, \{\tau_i\}_{i=1}^G \sim \pi_\theta^{\text{old}}(\cdot|q)} \left[ \frac{1}{\sum_{i=1}^G \sum_{t=1}^{|\tau_i|} \mathbb{I}(\tau_{i,t})} \sum_{i=1}^G \sum_{t=1}^{|\tau_i|} \mathbb{I}(\tau_{i,t}) \cdot \min\left( \frac{\pi_\theta(\tau_{i,t}|\tau_{i,<t})}{\pi_\theta^{\text{old}}(\tau_{i,t}|\tau_{\tau,<t})} \hat{A}_{i,t}, \right. \right.$$
$$\left. \left. \text{clip}\left( \frac{\pi_\theta(\tau_{i,t}|\tau_{i,<t})}{\pi_\theta^{\text{old}}(\tau_{i,t}|\tau_{i,<t})}, 1 - \epsilon_{\text{low}}, 1 + \epsilon_{\text{high}} \right) \hat{A}_{i,t} \right) \right], \quad \text{s.t.} \ 0 < \sum_{i=1}^G R_i < G \tag{2}$$

where the indicator $\mathbb{I}(\tau_{i,t}) = 1$ if $\tau_{i,t}$ is a generated token; and 0 otherwise. The advantage of the $i$-th response is estimated as: $\hat{A}_{i,t} = [R_i - \text{mean}(\{R_i\}_{i=1}^G)]/\text{std}(\{R_i\}_{i=1}^G)$. As a result, we obtain the policy model `control-32b-rl`. Table 14 (§ H) lists the hyperparameters for DAPO.

## 5 EXPERIMENTS

### 5.1 BASELINES

We evaluate M3-Agent against three types of baselines: (1) **Socratic Models**: A class of modular frameworks (Zeng et al., 2022) that first use an MLLM to generate textual descriptions for

each video clip and store them as long-term memory. When answering questions, an LLM retrieves these descriptions via a `search_clip` function and performs reasoning through RAG. We implement several variants with different MLLMs for description: proprietary models including Gemini-1.5-Pro (Team et al., 2024) and GPT-4o (Hurst et al., 2024), and open-source models including Qwen2.5-Omni-7b (Xu et al., 2025a) and Qwen2.5-VL-7b (Bai et al., 2025). GPT-4o is used uniformly for the final RAG step. (2) **Online video understanding methods**: Representative online video understanding methods, including MovieChat (Song et al., 2024), MA-LMM (He et al., 2024a), and Flash-VStream (Zhang et al., 2024b). (3) **Agent Methods**: Naive implementation of M3-Agent by prompting proprietary models to perform memorization and control. We implement Gemini-Agent, which uses Gemini-1.5-Pro for both memorization (`memory-gemini-prompt`) and control (`control-gemin-prompt`), and Gemini-GPT4o-Hybrid, which uses GPT-4o for control (`control-gpt4o-prompt`) and remains memorization handled by `memory-gemini-prompt`. We set the maximum number of execution rounds $H = 5$ for M3-Agent and all agent-based baselines, and configure `search_clip` to return the top 2 most relevant clips. Further implementation details of all baselines are provided in Appendix I.

## 5.2 DATASET AND EVALUATION

We evaluate M3-Agent and all baselines on both M3-Bench-robot and M3-Bench-web. To demonstrate the generality of our approach, we also test them on VideoMME-long (Fu et al., 2025), a mainstream long-video understanding benchmark, following its official evaluation protocol[†].

Table 4: Results on M3-Bench-robot, M3-Bench-web, and VideoMME-long, with comparisons across M3-Bench question types: multi-evidence reasoning (ME), multi-hop reasoning (MH), cross-modal reasoning (CM), person understanding (PU), and general knowledge extraction (GK).

| Method | M3-Bench-robot | | | | | | M3-Bench-web | | | | | | Video-MME-Long |
|---|---|---|---|---|---|---|---|---|---|---|---|---|---|
| | ME | MH | CM | PU | GK | All | ME | MH | CM | PU | GK | All | |
| *Socratic Model* | | | | | | | | | | | | | |
| Qwen2.5-Omni-7b | 2.1 | 1.4 | 1.5 | 1.5 | 2.1 | 2.0 | 8.9 | 8.8 | 13.7 | 10.8 | 14.1 | 11.3 | 42.2 |
| Qwen2.5-VL-7b | 2.9 | 3.8 | 3.6 | 4.6 | 3.4 | 3.4 | 11.9 | 10.5 | 13.4 | 14.0 | 20.9 | 14.9 | 46.9 |
| Gemini-1.5-Pro | 6.5 | 7.5 | 8.0 | 9.7 | 7.6 | 8.0 | 18.0 | 17.9 | 23.8 | 23.1 | 28.7 | 23.2 | 38.0 |
| GPT-4o | 9.3 | 9.0 | 8.4 | 10.2 | 7.3 | 8.5 | 21.3 | 21.9 | 30.9 | 27.1 | 39.6 | 28.7 | 38.8 |
| *Online Video Understanding Methods* | | | | | | | | | | | | | |
| MovieChat | 13.3 | 9.8 | 12.2 | 15.7 | 7.0 | 11.2 | 12.2 | 6.6 | 12.5 | 17.4 | 11.1 | 12.6 | 19.5 |
| MA-LMM | 25.6 | 23.4 | 22.7 | 39.1 | 14.4 | 24.4 | 26.8 | 10.5 | 22.4 | 39.3 | 15.8 | 24.3 | 17.3 |
| Flash-VStream | 21.6 | 19.4 | 19.3 | 24.3 | 14.1 | 19.4 | 24.5 | 10.3 | 24.6 | 32.5 | 20.2 | 23.6 | 25.0 |
| *Agent Method* | | | | | | | | | | | | | |
| Gemini-Agent | 15.8 | 17.1 | 15.3 | 20.0 | 15.5 | 16.9 | 29.3 | 20.9 | 33.8 | 34.6 | 45.0 | 34.1 | 55.1 |
| Gemini-GPT4o-Hybrid | 21.3 | 25.5 | 22.7 | 28.8 | **23.1** | 24.0 | 35.9 | 26.2 | 37.6 | 43.8 | 52.2 | 41.2 | 56.5 |
| **M3-Agent** | **32.8** | **29.4** | **31.2** | **43.3** | 19.1 | **30.7** | **45.9** | **28.4** | **44.3** | **59.3** | **53.9** | **48.9** | **61.8** |

## 5.3 MAIN RESULTS

As shown in Table 4, M3-Agent outperforms all baselines on M3-Bench-robot, M3-Bench-web, and VideoMME-long. Specifically, on M3-Bench-robot, M3-Agent achieves a 6.3% accuracy improvement over the strongest baseline, MA-LLM. On M3-Bench-web and VideoMME-long, it surpasses the strongest baseline, Gemini-GPT4o-Hybrid, by 7.7% and 5.3%, respectively.

Table 4 also reports a breakdown of performance across different question types in M3-Bench. M3-Agent shows strong performance in person understanding and cross-modal reasoning. Specifically, compared to the best-performing baseline on M3-Bench-robot, MA-LMM, M3-Agent achieves improvements of 4.2% in person understanding and 8.5% in cross-modal reasoning. On M3-Bench-web, M3-Agent outperforms the top baseline, Gemini-GPT4o-Hybrid, with gains of 15.5% and 6.7% in the respective categories. These results demonstrate M3-Agent 's superior ability to maintain character consistency, deepen person understanding, and integrate multimodal information.

We also assess the memorization model via precision and comprehension in Appendix G.

## 5.4 ABLATION STUDY

To evaluate the impact of memorization on performance, we fix the control model to `control-32b-rl` and compare different memorization methods (Table 5). Replacing the

---

[†]`https://github.com/thanku-all/parse_answer/blob/main/eval_your_results.py`

Table 5: Ablation of memorization models. The control is fixed as `control-32b-rl`.

| Memorization Model | M3-Bench-robot | M3-Bench-web | Video-MME-Long |
|---|---|---|---|
| `memory-gemini-prompt` | 28.7 | 46.3 | 52.7 |
| `memory-7b-prompt` | 25.3 | 39.9 | 50.8 |
| `memory-7b-sft` (M3-Agent) | **30.7** | **48.9** | **61.8** |
| `memory-7b-sft` w/o equivalence | 19.5 | 39.7 | 52.1 |
| `memory-7b-sft` w/o semantic memory | 13.6 | 29.7 | 48.7 |

Table 6: Ablation of control methods, including: (1) a comparison between GRPO and DAPO training algorithms; (2) performance gains from DAPO scale with model size; (3) the effect of removing inter-turn instruction and reasoning. The memorization model is fixed as `memory-7b-sft`.

| Control Model | M3-Bench-robot | M3-Bench-web | Video-MME-Long |
|---|---|---|---|
| `control-32b-grpo` | 30.0 | 47.7 | 58.7 |
| `control-8b-prompt` | 16.4 | 35.7 | 45.3 |
| `control-8b-rl` | 24.6 | 40.5 | 50.8 |
| `control-14b-prompt` | 18.3 | 36.9 | 49.1 |
| `control-14b-rl` | 28.2 | 46.9 | 56.0 |
| `control-32b-prompt` | 20.7 | 40.9 | 52.5 |
| `control-32b-rl` (M3-Agent) | **30.7** | **48.9** | **61.8** |
| `control-32b-prompt` w/o inter-turn instruction | 12.8 | 32.3 | 48.3 |
| `control-32b-rl` w/o inter-turn instruction | 20.2 | 43.1 | 55.9 |
| `control-32b-rl` w/o reasoning | 19.0 | 40.1 | 52.3 |

memory with that from `memory-gemini-prompt` reduces accuracy by 2.0%, 2.6%, and 9.1% on M3-Bench-robot, M3-Bench-web, and VideoMME-long, respectively, indicating that `memory-7b-sft` produces higher-quality memory than `memory-gemini-prompt`. Using `memory-7b-prompt` leads to accuracy reductions of 5.4%, 9.0%, and 11.0% on the same benchmarks, highlighting the importance of imitation learning in generating useful memory. Finally, removing character identity equivalence or semantic memory significantly degrades QA performance.

Next, we examine the impact of control on final performance, by fixing the memorization model to `memory-7b-sft` and evaluating various control models (Table 6). First, we compare two RL algorithms: GRPO and DAPO (training details for GRPO in Appendix H). Our results show that `control-32b-rl` trained with DAPO consistently outperforms `control-32b-grpo` across all test sets. Second, we analyze DAPO's scalability and observe substantial gains at all model sizes. Specifically, `control-32b-rl` achieves improvements of 10.0%, 8.0%, and 9.3% in accuracy over `control-32b-prompt` on M3-Bench-robot, M3-Bench-web, and VideoMME-long, respectively. Finally, ablation studies reveal that both inter-instruction and reasoning are critical: removing inter-instruction decreases accuracy by 10.5%, 5.8%, and 5.9%, while removing reasoning reduces accuracy by 11.7%, 8.8%, and 9.5% on the same benchmarks.

## 6 CONCLUSION

In this paper, we have introduced M3-Agent, a multimodal agent framework equipped with long-term memory. M3-Agent perceives real-time video and audio streams to build both episodic and semantic memories, enabling it to accumulate world knowledge and maintain consistent and context-rich memory over time. When given instructions, M3-Agent can autonomously reason and retrieve relevant memories to complete tasks more effectively. To evaluate memory effectiveness and reasoning capabilities, we have developed M3-Bench, an LVQA benchmark featuring real-world robot-perspective videos and challenging questions requiring person understanding, knowledge extraction, and cross-modal reasoning. Experiments on M3-Bench-robot, M3-Bench-web, and VideoMME-long show that M3-Agent consistently outperforms baselines, including Socratic models, online video understanding methods, and agents implemented by prompting closed-source models, demonstrating its superior memorization and reasoning capabilities. Finally, our case studies (Appendix L) reveal key limitations and promising future directions. These include enhancing attention mechanisms for semantic memory formation and developing richer yet more efficient visual memory.

## ETHICS STATEMENT

This research aims to advance the development of multimodal agents with human-like long-term memory capabilities. For the M3-Bench-robot subset, all actors were fully informed about the purpose of the recordings and signed explicit consent agreements outlining data usage. As for the M3-Bench-web subset, only publicly available videos were used, and annotations were created to ensure privacy and fairness. Therefore, the datasets used and released do not involve any privacy violations or ethical risks.

## REPRODUCIBILITY STATEMENT

To support transparency and reproducibility, we commit to release M3-Bench benchmark, model checkpoints, training data, and code, upon publication. The M3-Bench will include all robot-perspective and web-sourced videos along with detailed question–answer annotations and evaluation scripts. We will provide the model checkpoints for both memorization (`memory-7b-sft`) and control (`control-32b-rl`) described in the paper. The code release will cover memorization and control pipelines, tool implementations, and demonstration data synthesis pipelines. These resources will enable researchers to reproduce our experimental results and extend the work in future directions.

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

# Appendix

## A  LLM USAGE DISCLOSURE

We used large language models (LLM) for the language-related aspects of this paper, including spell-checking, grammar-checking, and refining our original text. No new content was generated by the LLMs. All LLM-refined text was either carefully reviewed or further edited by the authors. Additionally, we employed an LLM-based paper search agent[†] to complement, but not replace, our manual survey of related works.

## B  M3-BENCH-ROBOT ANNOTATION DETAILS

### B.1  SCRIPT ANNOTATION GUIDELINES

**Actor Setup**

Four to five actors participate, including one playing the role of robot.

**Definitions**

1. Script: Consists of events and questions and provides actors with dialogue and stage instructions.

2. Robot: Played by a human actor. It is an ideal highly intelligent robot with reasoning and memory abilities similar to humans.

3. Scenario: living room, kitchen, bedroom, study, office, meeting room, and gym.

4. Event: A complete, short plot within the script. A *reference event* includes information relevant to future questions, such as robots interacting with humans while observing and learning human preferences or the placement of objects in real-world scenes.

5. Question: Designed to evaluate the robot's memory. Each question must align with at least one type listed in Table 1.

**Requirements**

- Annotate at least 15 questions, each labeled with the corresponding reference events.
- Each script must contain at least 70 events to ensure a minimum video duration of 30 minutes.
- Avoid asking questions that rely solely on common sense or that can be answered without watching the video.
- Do not ask questions that remain unanswerable even after watching the video.
- Avoid questions that can be answered based solely on the dialogue.
- Do no include questions that are weakly related to the reference events.
- The question should have a clear and unambiguous answer that can be objectively verified by comparing it to the reference answer.

### B.2  VIDEO FILMING

The robot actor wears a head-mounted camera, either an iPhone 16 Pro, Xiaomi 14 Ultra, or GoPro HERO13, to capture a single point-of-view video from the robot's perspective.

We collect two types of audio tracks for each video. The first is directly recorded by the head-mounted device, reflecting the raw auditory input a robot would naturally receive, including ambient sounds and spatial acoustic variations. The second is captured using individual lapel microphones worn by each actor, providing high-fidelity voice recordings to complement the primary audio stream.

### B.3  QA ANNOTATION GUIDELINES

**Background**

---

[†]https://pasa-agent.ai/

- In the future, robots will help humans complete many tasks in indoor environments such as homes. Based on this imagination, we filmed a video from the perspective of a robot.

- In order to evaluate the model's ability, we set questions at different timestamps, typically related to the robot's upcoming tasks. Correct answers are essential for the successful completion of these tasks.

- Some questions require manual review or additional annotations to ensure each video includes at least 10 questions.

**Task**

Provide a 30–45 minute video along with a corresponding script that includes a series of questions. Note: Minor script modifications may occur during filming to accommodate practical constraints. As a result, the script may not perfectly align with the final video.

1. Review existing questions.

For each question in the script:

- Annotate the corresponding timestamp in the video based on the related script event.

- Determine whether the question can be answered using the video content up to that point. If so, annotate the answer.

- If the question is unanswerable, consider whether modifying it could make it answerable. If applicable, revise the question and provide the answer.

- For each question-answer pair, annotate the reasoning process used to derive the answer and specify the question types according to Table 1.

2. Annotate additional questions:

If fewer than 10 questions remain after reviewing the script, generate new questions that must belong to at least one type listed in Table 1.

### B.4    QUALITY CONTROL

The annotation process consists of two rounds. In the first round, the goal is to ensure that annotators fully understand the annotation guidelines. Each annotator is required to perform QA annotations on three videos. The authors then review the annotations, provide feedback, and the annotators may revise their annotation accordingly. Based on the quality of these initial annotations, the authors determine whether the annotator is qualified to proceed to the formal annotation phase. In the second round, each annotator annotates five videos at a time. The authors randomly select one video from each batch for quality inspection. If more than one invalid question-answer is found in the selected video, the entire batch must be re-annotated. Otherwise, the batch is considered accepted. Two authors are involved in the quality control process throughout the annotation workflow.

In addition, to ensure the quality of the questions in M3-Bench-robot, we recruit five annotators to answer each question. Annotators are allowed to first read the question and then watch the video as many times as needed. The final human accuracy on M3-Bench-robot is 90.7%. Our error analysis shows that the most common mistakes are counting-related problems.

### B.5    ANNOTATOR INFORMATION

All annotators are employed by a commercial data annotation company. We sign a contract with the company and pay the company for the annotation work at a market price. The annotators are all college graduates with strong English proficiency. For script annotation, eleven annotators are involved. Video filming engage 67 actors. For QA annotation, five annotators participate.

### B.6    DATA EXAMPLE

Table 7 provides an example of script annotation.

| Event ID | Event |
|---|---|
| 1 | Rose is in the room talking to Amy on the phone. She thanks Amy for the tulips and takes a photo of the blooming flowers to share with her. *(reference)* |
| 2 | Rose tells the robot that the delicate teddy bear is a gift for Rachel. *(reference)* |
| 3 | After hanging up with Amy, Rose calls Rachel and Leo to remind them not to forget to come over today. |
| 4 | Rose looks at a pile of packages in the corner of the bedroom. They are recently purchased clothes. She asks the robot to unpack them and place the clothes on the first shelf of the wardrobe. |
| 5 | She points to the bottom of the wardrobe, where a pile of delicate little toys is stored, and tells the robot, "Put the teddy bear there." *(reference)* |
| 6 | At that moment, the doorbell rings and Rose excitedly runs to open the door. |
| ... | ... |
| 10 | Rachel sees the dolls on the bed and exclaims, "Wow, these dolls are so cute, let me pamper them!" |
| 11 | Rose says, "Don't rush, there's another surprise," and then calls the robot. **Question:** Is Rachel's gift on the top shelf or the bottom shelf of the wardrobe? **Reference:** event-2 and event-5 |
| 12 | The robot takes a teddy bear from the wardrobe, hands it to Rachel, and says, "This is a gift prepared for you." |
| ... | ... |
| 58 | Rachel teases that Rose just doesn't want to admit it, but the robot surely knows. She then turns to the robot and asks who gave Rose the flowers. **Question:** Who gave Rose the flowers? **Reference:** event-1 |
| ... | ... |

Table 7: An example of the M3-Bench-robot script.

## C   M3-BENCH-WEB ANNOTATION DETAILS

### C.1   ANNOTATION GUIDELINES

To better help the annotators understand the requirements and better ensure the overall quality, safety, and validity of the datasets, we provide the following detailed guidelines, which clearly specify the acceptable and unacceptable annotation practices.

- **Questions must allow for verifiable and objective evaluation of correctness**. This entails avoiding overly open-ended questions, compound questions that mix multiple sub-questions, or questions with multiple equally valid answers.

- **Each video must include at least two questions targeting character attribute modeling and two questions involving commonsense reasoning**.

- **All visual information required to answer a question must remain clearly recognizable at lower resolutions ($\leq$720p)**, ensuring that all questions are answerable.

- **For videos between 20 and 40 minutes in length, 5 questions should be generated; for videos exceeding 40 minutes, 10 questions should be provided**. Compensation considers both the number and duration of the videos.

- **For commonsense reasoning questions, annotators must also specify the commonsense knowledge being tested**, in addition to the question and its answer.

- **It is not permissible for all questions to be answerable using only audio**. A reasonable proportion of questions must be vision-centric, requiring understanding of visual content in the video.

- **Redundant questions within the same video are not allowed**. For instance, asking "Describe David's appearance" and "Describe Alice's appearance" would be considered repetitive.

- **Questions that can be answered solely based on a brief moment or a short clip should be avoided**. Specifically, the context required to answer a valid question should span more than 10 seconds of video content.

- **Videos must not contain sensitive, offensive, or NSFW content**.

- **Avoid asking questions that rely solely on commonsense knowledge and do not require viewing the video**. Such questions do not meaningfully test video understanding.

- **Avoid questions that are too easy to guess based on social priors or language bias alone**. For example, a question like "Did the teacher appear impatient when students repeatedly interrupted the class?" may be too easily answered with "No" due to cultural expectations of teacher behavior, regardless of the actual video content. This undermines the goal of evaluating visual understanding.

- **Do not directly convert characters' spoken lines into questions**. These are typically answerable via simple string matching or keyword retrieval, which again does not effectively test video comprehension.

- **Balance the number of questions with answer "Yes" and "No"**.

## C.2 QUALITY CONTROL

The annotation process includes the following quality control stages:

- Stage 1: Candidate annotators complete a trial task, collecting one video and labeling corresponding QA pairs. The authors review the submission and provide feedback. Once the annotator demonstrates a clear understanding of the annotation guidelines, they proceed to formal annotation.

- Stage 2: The annotator submits a batch of 10 videos with corresponding QA pairs. The authors randomly review 2 of them and provide feedback. The annotator revise the entire batch accordingly. If the qualified rate of the submitted questions is below 90%, the authors re-sample the revised batch for further inspection. Otherwise, the batch is accepted. Annotators who pass this stage on the first attempt can proceed to Stage 3.

- Stage 3: The annotator submits a batch of 30 videos with QA pairs. The authors randomly inspect 5 of them and provide feedback. The annotator revises the full batch as needed. If the QA qualified rate is below 90%, a follow-up review of the revised batch is conducted. Otherwise, the batch is accepted.

Two authors are involved in the quality control process.

## C.3 ANNOTATOR INFORMATION

All annotators are from a commercial data annotation company. We have a contract with this company and compensate them at market rates for the annotation work. All annotators are college graduates with strong English proficiency. A total of ten annotators participate in the annotation of M3-Bench-web.

## D LONG-TERM MEMORY DESIGN

The long-Term memory of M3-Agent is implemented as an entity-centric, multimodal graph. Each node represents a distinct memory item, and each edge represents logical relationships between memory items. The attributes associated with each memory node are listed in Table 8.

Table 8: Attributes and their descriptions for a memory node.

| Attribute | Description |
|---|---|
| id | A unique identifier for the node. |
| type | The modality type of the node (e.g., text, image, audio). For example, natural language memory is stored as a text node, a face as an image node, and spoken dialogue as an audio node. |
| content | The raw content of the node, such as plain text, base64 image, or base64 audio. |
| embedding | The vector representation of the node content, used for similarity-based retrieval. |
| weight | A numeric value indicating the confidence of the node. |
| extra_data | A JSON object containing additional metadata, such as timestamps. |

Conflicting information may be introduced during memory construction. To address this, M3-Agent applies a weight-based voting mechanism, where weight of each entry reflects its activation frequency. For example, `<voice_3>` corresponds to `<face_0>`, but in some challenging clips, the system might temporarily link it to a different face. Over time, as the correct association is activated more frequently, the weight of the correct mapping (`<voice_3>`, `<face_0>`) increases and dominates. This allows the system to robustly learn and maintain accurate knowledge, even in the presence of occasional local errors.

In long-term memory, M3-Agent can associate multimodal information, even across different time steps. For instance, when a voice appears without a face, M3-Agent generates the related episodic and semantic memories with the corresponding voice ID (e.g., `<voice_0>`). If the corresponding face (e.g. `<face_1>`) is observed in any past or future, M3-Agent can associate the two modalities. Consequently, all knowledge related to `<voice_0>` can be also accessed by `<face_1>`.

## E    IMPLEMENTATION DETAILS OF TOOLS

Here, we provide the implementation details of the tools for representation extraction introduced in Section 4.2.

**Facial Recognition** To perform facial recognition, we uniformly sample video frames at a rate of 5 frames per second. For each sampled frame, we employ the `buffalo_l` predefined model suite from the `InsightFace`[†] library to extract facial attributes, including bounding box coordinates, identity embeddings, and detection/quality scores. Low-quality detections—such as those with abnormal aspect ratios or extremely low confidence scores—are discarded. We then apply `HDBSCAN` clustering on the embeddings of the remaining high-quality faces to group them by character identity. This yields a set of reliable facial representations, clustered by character.

**Voice Identification** For speaker identification, we use Gemini-1.5-Pro to extract audio segments corresponding to distinct speaker voices, while simultaneously performing automatic speech recognition (ASR) on each segment. Segments shorter than 2 seconds are filtered out to ensure reliability. We then apply voice embedding model ERes2NetV2(Chen et al., 2024a) to encode each segment into a speaker-specific representation. Based on the resulting voice embeddings, we cluster and merge segments that correspond to the same speaker—i.e., those with similar vocal characteristics. This process produces a set of high-quality speaker representations, also grouped by character. The prompt used for voice processing is shown in Table 9.

---

**The Prompt for Voice Processing**

You are given a video. Your task is to perform Automatic Speech Recognition (ASR) and audio diarization on the provided video. Extract all speech segments with accurate timestamps and segment them by speaker turns (i.e., different speakers should have separate segments), but without assigning speaker identifiers.

---

*(Continued on next page)*

[†]`https://github.com/deepinsight/insightface`

Return a JSON list where each entry represents a speech segment with the following fields:
• start_time: Start timestamp in MM:SS format.
• end_time: End timestamp in MM:SS format.
• asr: The transcribed text for that segment.

Example Output:
[
    "start_time": "00:05", "end_time": "00:08", "asr": "Hello, everyone.",
    "start_time": "00:09", "end_time": "00:12", "asr": "Welcome to the meeting."
]

Strict Requirements:
• Ensure precise speech segmentation with accurate timestamps.
• Segment based on speaker turns (i.e., different speakers' utterances should be separated).
• Preserve punctuation and capitalization in the ASR output.
• Skip the speeches that can hardly be clearly recognized.
• Return only the valid JSON list (which starts with "[" and ends with "]") without additional explanations.
• If the video contains no speech, return an empty list ("[]").

Now generate the JSON list based on the given video:

Table 9: Prompt used for voice processing.

**Search** All memory-based retrieval is implemented via Maximum Inner Product Search (MIPS), with modality-specific adaptations.

Each face and voice node maintains a set of representative feature snapshots. When new face or voice features are extracted from a video clip, we compute the average cosine similarity between each extracted feature and all stored snapshots per node. The node with the highest similarity exceeding a pre-defined threshold (0.3 for image, 0.6 for voice) is considered a match; otherwise, a new node is created. Matched nodes are updated with the new features to refine their representations over time.

For textual memory, we apply MIPS between the input query and all existing text nodes, using OpenAI's `text-embedding-3-large`[†] as the embedding model. To support multi-entry retrieval, we apply a top-$k$ retrieval with a similarity threshold $t$. Specifically, we return the $k$ most relevant nodes whose similarities exceed $t$. To ensure retrieval coherence, we also perform clip-level retrieval: each clip is scored by the highest similarity among its memory entries, and we return the top-ranked clips accordingly. For all experiments, we adopt a relatively strict hyperparameter setting ($k = 2$, $t = 0.5$) to reduce retrieval randomness and enable consistent evaluation across models.

## F    DEMONSTRATION DATA SYNTHESIS FOR MEMORIZATION

During memorization, the multimodal model takes inputs including: video, audio, facial identifications (via facial recognition), and voice identities (via voice identification). It generates two outputs, episodic memory and semantic memory. To construct training data, we segment training videos into 30-second clips. For each clip, we then synthesize the corresponding episodic memory, entity identity relationships in semantic memory, and other semantic memory, as detailed below. In total, we synthesize 10,752 training samples for 200 validation samples.

### F.1    EPISODIC MEMORY SYNTHESIS

We employ a hybrid synthetic strategy that integrates the complementary strengths of Gemini-1.5-Pro and GPT-4o. Gemini-1.5-Pro supports audio inputs and excels at generating high-level, event-based descriptions, whereas GPT-4o provides more fine-grained visual details. To leverage both

---

[†]https://openai.com/index/new-embedding-models-and-api-updates/

Table 10: Explanations of different memory types.

| Memory Type | Explanation |
|---|---|
| Episodic Memory | Specific events or experience, capturing not just what happened, but also when, where, and in what context. The episodic memory should captures details such as the people involved, their appearance, actions and spoken words, and the broader environment. |
| Semantic Memory | • *Character-Identity Equivalence*: Captures equivalence relationships across different character modality identity
• *Character-Level Attributes*: Extracts attributes for each character, such as name, personality traits (e.g., confident, nervous), role or profession (e.g., host, newcomer), interests, and background information.
• *Interpersonal Relationships*: Describes the relationships and interactions among characters, such as social roles (e.g., host–guest, leader–subordinate), emotional tone (e.g., respect, tension), power dynamics (e.g., who leads), and evidence of cooperation, exclusion, or conflict.
• *Contextual and General Knowledge*: Encompasses general knowledge inferred from the video, such as likely setting or genre (e.g., corporate meeting, game show), cultural or procedural norms, real-world facts (e.g., "Alice Market is pet-friendly"), common sense, and the functional roles or attributes of objects within the scene. |

models effectively, we first prompt GPT-4o to generate a detailed visual description of the video using frames sampled at 0.5 fps. This output serves as contextual input for Gemini-1.5-Pro, which is then prompted to generate the final episodic memory. The prompt explicitly instructs Gemini-1.5-Pro to incorporate information from GPT-4o's description when it deems it accurate. We find that using GPT-4o's detailed visual output as context significantly enhances the richness of the final memory produced by Gemini-1.5-Pro. The full prompt template is shown in Table 11.

---

**Prompt of Episodic Memory Synthesis (GPT-4o)**

[Video] includes 16 frames of a video.

Using this information, generate a detailed description of the video. Following the requirements below:
1. Carefully describe the visual elements in each frame, noting colors, objects, movements, environment, people (including actions, clothing, expressions), and any noticeable details or changes between frames.
2. If audio elements or sounds are visible through textual or visual cues within the frames (such as subtitles, audio indicators, or written sound effects), accurately describe these details.
3. Do not speculate or infer information beyond what is explicitly visible in these 16 frames. Avoid using external knowledge or assumptions.
4. Generate only the detailed description based solely on the given frames. Do not produce any additional commentary or explanations.

**Prompt of Episodic Memory Synthesis (Gemini-1.5-Pro)**

You are provided with the following data:
[Video]: A video clip in mp4 format.
[Faces]: A list of facial features detected in the video, each linked to a unique face ID (e.g., <face_1>).
[Dialogues]: A list of speech segments in the video, including start_time, end_time, speaker ID (e.g., <voice_2>), and the corresponding transcribed text.
[Reference Description]: A description of the video that may contain both accurate and inaccurate details.

Your Tasks:

Based on the video content and reference descriptions, generate a detailed and cohesive description of the video clip. The description should focus on the entire event, incorporating all relevant aspects of the characters, their actions, spoken dialogue, and interactions in a narrative format. The description should include (but is not limited to) the following categories:
• Characters' Appearance: Describe clothing, physical features, notable accessories, etc.
• Characters' Actions & Movements: Describe gestures, movement across the scene, or interactions.
• Characters' Spoken Dialogue: Quote—or, if necessary, summarize—spoken content from the dialogue track.
• Characters' Contextual Behavior: Describe emotional states, relationships, roles, and reactions.
• Environmental or Temporal Cues: Describe the physical setting and time-of-day if visible.

Strict Requirements:
• Incorporate correct elements from the [Reference Description], and correct any mistakes you identify.
• Add any additional details visible or inferable from the [Video], [Faces], and [Dialogues] that are missing from the reference.
• Since the given dialogues may be incomplete, reconstruct the entire conversation from the raw audio as precisely as possible.
• If a character has an associated feature ID in the input context (either face or voice), refer to them only using that feature ID (e.g., <face_1>, <voice_2>)
    • Use face ID (e.g., <face_1>) when the detail is grounded in visual data.
    • Use speaker ID (e.g., <voice_1>) when the detail is grounded in speech.
• Do not use non-existent <face_ID> or <voice_ID>.
    • We reiterate the above-mentioned list of available IDs here: {ID_list}
• For characters without associated feature IDs, refer to them using a concise visual or contextual descriptor (e.g., "a man in a blue shirt", "a young woman by the window").
• Do not use pronouns (e.g., "he", "she", "they") or inferred character names.

Your output should be a Python list of well-formed, concise English sentences (one detail per sentence).

Example Output:
[
    "In the bright conference room, <face_1> enters confidently, adjusting his black suit with a white shirt and tie. He has short black hair and wears glasses, giving a professional appearance as he approaches <face_2> to shake hands.",
    "<face_2>, dressed in a striking red dress with long brown hair, smiles warmly and greets <face_1>. She then sits down at the table beside him, glancing at her phone briefly while occasionally looking up.",
    "<voice_1> speaks to the group, 'Good afternoon, everyone. Let's begin the meeting.' His voice commands attention as the room quiets, and all eyes turn to him.",
    "<face_2> listens attentively to <voice_1>'s words, nodding in agreement while still occasionally checking her phone. The atmosphere is professional, with the participants settling into their roles for the meeting.",
    "<face_1> adjusts his tie and begins discussing the agenda, engaging the participants in a productive conversation."
]

Please only return the valid string list (which starts with "[" and ends with "]"), without any additional explanation or formatting.

Table 11: Prompt templates used for generating synthetic episodic memory.

F.2   ENTITY ID RELATIONSHIP DETECTION

There is a special type of semantic memory, extracting cross-modal identity equivalences from video. This remains a challenging task, even for advanced models like Gemini-1.5-Pro, particularly in scenes with multiple faces and voices (He et al., 2024b). To address this, we propose a progressive annotation algorithm. The key idea is to identify **meta-clips**, segments containing exactly one face identity and one voice identity, from the raw long video. These meta-clips are used to build a meta-dictionary that maps voice IDs to face IDs across the entire video. This dictionary enables automatic annotation of any 30-second clip extracted from the original video.

**Meta-Clip Extraction** First, for a long video, we can use facial recognition tools and voice identity tools introduced in Appendix E to construct a corresponding global ID for each face and voice that appears in the video. Next, we segment the video into a series of short clips, each no longer than 5 seconds in duration, using keyframe-based division. This method ensures that each clip is visually stable, with minimal changes in characters or scenes. Then, we apply facial recognition and voice identity tools to each short clip individually to extract the faces and voices present, along with their global IDs. If a clip contains only one face ID and one voice ID, we refer to it as a meta-clip. In this case, it is highly likely that the face and voice in the clip belong to the same person. Therefore, we can use the meta-clip as a high-confidence sample for establishing the association between faces and voices.

---

**Algorithm 2** Meta-Dictionary Construction

---

**Require:** A long video $V$, threshold $p$
**Ensure:** A mapping dictionary $\mathcal{M} : \mathcal{V} \to \mathcal{F}$ from voice IDs to face IDs
 1: Extract global face ID set $\mathcal{F} = \{f_1, \ldots, f_N\}$ and voice ID set $\mathcal{V} = \{v_1, \ldots, v_N\}$ from video $V$
 2: Divide $V$ into a sequence of short clips $\mathcal{C} = \{c_1, c_2, \ldots, c_T\}$ using keyframes-based segmentation
 3: Initialize meta-clip set $\mathcal{C}_{\text{meta}} \leftarrow \emptyset$
 4: **for** $c_t \in \mathcal{C}$ **do**
 5:      Detect face set $\mathcal{F}_t \subseteq \mathcal{F}$ and voice set $\mathcal{V}_t \subseteq \mathcal{V}$ in $c_t$
 6:      **if** $|\mathcal{F}_t| = 1$ **and** $|\mathcal{V}_t| = 1$ **then**
 7:          Add pair $(c_t, f, v)$ where $f \in \mathcal{F}_t$, $v \in \mathcal{V}_t$ to $\mathcal{C}_{\text{meta}}$
 8:      **end if**
 9: **end for**
10: Construct bipartite graph $G = (\mathcal{F}, \mathcal{V}, E)$ where edge $(f, v)$ has weight: $w(f, v) = |\{(c_t, f, v) \in \mathcal{C}_{\text{meta}}\}|$
11: Remove all edges from $G$ with weight equal to 1.
12: **for** $f \in \mathcal{F}$ **do**
13:      Let $\mathcal{N}_f = \{v_i \mid (f, v_i) \in E\}$
14:      Let $v^* = \arg\max_{v_i \in \mathcal{N}_f} w(f, v_i)$
15:      **if** $\frac{w(f,v^*)}{\sum_{v_i \in \mathcal{N}_f} w(f,v_i)} \geq p$ **then**
16:          Keep only edge $(f, v^*)$ and remove others
17:      **else**
18:          Remove all edges incident to $f$
19:      **end if**
20: **end for**
21: **for** $v \in \mathcal{V}$ **do**
22:      Let $\mathcal{N}_v = \{f_j \mid (f_j, v) \in E\}$
23:      Let $f^* = \arg\max_{f_j \in \mathcal{N}_v} w(f_j, v)$
24:      Keep only edge $(f^*, v)$ and remove others
25: **end for**
26: Initialize mapping dictionary $\mathcal{M} \leftarrow \emptyset$
27: **for** $(f, v) \in E$ **do**
28:      Add mapping $\mathcal{M}[v] \leftarrow f$
29: **end for**
30: **return** $\mathcal{M}$

---

**Meta-Dictionary Construction** Based on all meta-clips extracted from the long video, we construct a set of mappings between face IDs and voice IDs. However inconsistencies may arise due to a small number of clips where the speaker is not visible. To address this issue, we employ a voting mechanism to generate the final meta-dictionary. The detailed algorithm is described in Algorithm 2.

**New-Clip Annotation** After obtaining the meta-dictionary, we can use it to annotate arbitrary clips from the full-length video. Specifically, for each 30-second clip, if both a face ID and a voice ID appearing in the clip and also found in the meta-dictionary, we generate a semantic memory in the form: "Equivalence: `<face_id>`, `<voice_id>`". Since not all IDs can be found using the meta-dictionary, we reject any clip containing a voice ID that is not present in the meta-dictionary from the final training dataset for memorization. In total, we collected 10,952 30-second clips with valid identity equivalence annotations. We manually review 48 randomly sampled mappings, and found the accuracy to be 95.83%.

### F.3  SEMANTIC MEMORY SYNTHESIS

To construct semantic memory, we adopt a hybrid strategy similar to that used for episodic memory. We define several key dimensions that semantic memory should address, as outlined in Table 10. Specifically, we first prompt GPT-4o to generate preliminary semantic memory based on video frames and episodic memory. Next, we provide the video, episodic memory, and GPT-4o-generated semantic memory to Gemini-1.5-Pro, prompting it to produce the final semantic memory. Detailed prompts are provided in Table 12.

---

**Prompt of Semantic Memory Synthesis (GPT-4o, Gemini-1.5-Pro)**

You are provided with the following data:
[Video]: 16 frames of a video. (***Gemini-1.5-Pro Variant:*** *A video clip in mp4 format.*)
[Faces]: A list of facial features detected in the video, each linked to a unique face ID (e.g., `<face_1>`).
[Dialogues]: A list of speech segments in the video, including start_time, end_time, speaker ID (e.g., `<voice_2>`), and the corresponding transcribed text.
[Video Descriptions]: A description of the video.
(***Gemini-1.5-Pro Variant:*** [*Refence conclusions*]: *A list of high-level conclusions that may contain inadequate or incorrect information.*)

Your Task:

Based on the given character features, video content, and reference conclusions, generate a list of high-level, reasoning-based conclusions within the scope of the following category:

1. Character-Level Attributes
   Infer abstract attributes for each character, such as:
       • Name (if explicitly stated),
       • Personality (e.g., confident, nervous),
       • Role/profession (e.g., host, newcomer),
       • Interests or background (when inferable),
       • Distinctive behaviors or traits (e.g., speaks formally, fidgets).
   Avoid restating visual facts—focus on identity construction.

2. Interpersonal Relationships & Dynamics
   Describe the relationships and interactions between multiple characters:
       • Roles (e.g., host-guest, leader-subordinate),
       • Emotions or tone (e.g., respect, tension),
       • Power dynamics (e.g., who leads),
       • Evidence of cooperation, exclusion, conflict, etc.
       • For individual character or cases where character relationships cannot be determined, do not generate conclusion relevant to the corresponding character.

---

3. Video-Level Plot Understanding

Summarize the scene-level narrative, such as:
- Main event or theme,
- Narrative arc or sequence (e.g., intro → discussion → reaction),
- Overall tone (e.g., formal, tense),
- Cause-effect or group dynamics.
- Do not involve specific characters.

4. Contextual & General Knowledge

Include general knowledge that can be learned from the video, such as:
- Likely setting or genre (e.g., corporate meeting, game show),
- Cultural/procedural norms,
- Real-world knowledge (e.g., "Alice market is pet-friendly"),
- Common-sense or format conventions.
- Attributes and functional roles of objects in the video (e.g., the trash bin is used for disposing of kitchen waste).

Output Format:
- A Python list of concise English sentences, each expressing one high-level conclusion.
- Do not include reasoning steps or restate input observations. Only output the final conclusions.

Strict Requirements:
- Only include conclusions under the given category. Do not go beyond it.
- Your conclusions must be informed by the video and reference content.
- Each conclusion should reflect deeper reasoning and insight, not surface-level observations already evident from the plot description.
- If a character has an associated feature ID in the input context (either face or voice), refer to them only using that feature ID (e.g., <face_1>, <voice_2>).
    - Use face ID (e.g., <face_1>) when the detail is grounded in visual data.
    - Use speaker ID (e.g., <voice_1>) when the detail is grounded in speech.
- Do not use non-existent <face_ID> or <voice_ID>.
    - We reiterate the above-mentioned list of available IDs here: {ID_list}
- For characters without associated feature IDs, refer to them using a concise visual or contextual descriptor (e.g., "a man in a blue shirt", "a young woman by the window").
- Do not use pronouns (e.g., "he", "she", "they") or inferred character names.
- Maintain strict accuracy in referring to characters and their correct IDs or descriptions.
- Do not restate the input observations or reasoning steps—only output the final, distilled conclusions.
- Your output should be a Python list of well-formed, concise English sentences (one per item).

Example Output (Note: example only represent the format, not fully corresponding to the provided category):
[
    "<face_1>'s name is David.",
    "<face_1> holds a position of authority, likely as the meeting's organizer or a senior executive.",
    "<voice_2> shows social awareness and diplomacy, possibly indicating experience in public or client-facing roles.",
    "<face_1> demonstrates control and composure, suggesting a high level of professionalism and confidence under pressure.",
    "The interaction between <face_1> and <voice_2> suggests a working relationship built on mutual respect.",
    "The overall tone of the meeting is structured and goal-oriented, indicating it is part of a larger organizational workflow."
]

Please only return the valid string list (which starts with " [" and ends with "] ") without any additional explanation or formatting.

Table 12: The prompt used in generating synthetic semantic memory.

## F.4 QUALITY OF THE SYNTHETIC DATA

Although the demonstration data is synthetic, it is of high quality. Our synthetic memory averages 245.7 words for episodic memory and 276.2 words for semantic memory, compared to 151.3 and 81.4 words respectively for Gemini-1.5-pro, indicating our memory captures more detail. For content accuracy, we randomly sampled 10 clips from different videos, totaling 353 memory items. Manual review showed an accuracy of 95.5%. Most errors stemmed from the speaker recognition tool: background noise and overlapping speech occasionally caused minor omissions or misidentifications in extracting speaker dialogue for episodic memory.

## G EVALUATION OF MEMORIZATION

we evaluate the memorization model during training using a held-out validation set of 200 samples and select the best checkpoint. Two evaluation metrics are used. First, AutoDQ (Wang et al., 2024a) assesses memory description quality by comparing generated outputs to reference descriptions, measuring episodic and semantic memory excluding identity equivalence. Second, for identity equivalence, we compute precision, recall and F1 score against ground-truth in the validation set. Based on the results in Table 13, we select the checkpoint obtained after training for 3 epochs. For additional comparison, we also report results from two baseline models, `memory-gemini-prompt` and `memory-7b-prompt`, on the same validation set. Our model, `memory-7b-sft`, significantly outperforms both baselines.

Table 13: Evaluation of memorization models using AutoDQ and Equivalence (Eq.) metrics. Here, **P**, **R**, and **F1** denote precision, recall, and the F1 score, respectively.

| Model | AutoDQ-P | AutoDQ-R | AutoDQ-F1 | Eq.-P | Eq.-R | Eq.-F1 |
|---|---|---|---|---|---|---|
| memory-gemini-prompt | **0.692** | 0.539 | 0.606 | 0.472 | 0.805 | 0.595 |
| memory-7b-prompt | 0.495 | 0.355 | 0.414 | 0.117 | 0.192 | 0.145 |
| memory-7b-sft (1 epoch) | 0.634 | 0.596 | 0.616 | 0.742 | 0.817 | 0.778 |
| memory-7b-sft (2 epochs) | 0.628 | 0.610 | 0.619 | **0.845** | 0.810 | 0.827 |
| memory-7b-sft (3 epochs) | 0.635 | **0.620** | **0.627** | 0.836 | **0.856** | **0.846** |
| memory-7b-sft (4 epochs) | 0.616 | 0.618 | 0.617 | 0.825 | 0.839 | 0.832 |
| memory-7b-sft (5 epochs) | 0.609 | 0.621 | 0.615 | 0.813 | 0.840 | 0.827 |

## H RL TRAINING DETAILS

### H.1 DAPO TRAINING

Table 14 lists the hyperparameters used during the training process. Figure 4 depicts the RL training curves, which show a steady increase in score with the training steps.

### H.2 GRPO TRAINING

We also use Group Relative Policy Optimization (GRPO)(Shao et al., 2024) to optimize the policy model in the ablation study. GRPO optimizes the policy model $\pi_\theta$ by maximizing the following

Table 14: The hyperparameters used in DAPO training.

| Parameter Name | Model Size | | |
|---|---|---|---|
| | 8B | 14B | 32B |
| Batch Size | 32 | 32 | 32 |
| GPU with 80GB memory | 16 | 16 | 32 |
| Rollout Model Parallel Size | 1 | 1 | 2 |
| Learning Rate | 1e-6 | 1e-6 | 1e-6 |
| Maximum Number of Rounds $H$ | 5 | 5 | 5 |
| Number of Samples in a Group $G$ | 4 | 4 | 4 |
| Total Steps | 180 | 180 | 180 |
| $\epsilon_{\text{low}}$ | 0.2 | 0.2 | 0.2 |
| $\epsilon_{\text{high}}$ | 0.28 | 0.28 | 0.28 |

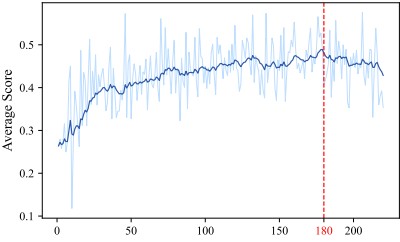 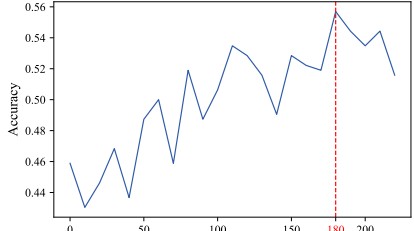

Figure 4: Average scores (on training set) and accuracy (on dev set) curves during the DAPO training process. The smoothing method of the curve in the left figure is the exponential moving average(EMA) formula that aligns with the one used in WandB, and the smoothing weight is set to 0.9

objective:

$$\mathcal{J}_{\text{GRPO}}(\theta) = \mathbb{E}_{(q,a)\sim\mathcal{D}, \{\tau_i\}_{i=1}^G \sim \pi_\theta^{\text{old}}(\cdot|q)} \left[ \frac{1}{G} \sum_{i=1}^G \frac{1}{\sum_{t=1}^{|\tau_i|} \mathbb{I}(\tau_{i,t})} \sum_{t=1}^{|\tau_i|} \mathbb{I}(\tau_{i,t}) \cdot \min \left( \frac{\pi_\theta(\tau_{i,t}|\tau_{i,<t})}{\pi_\theta^{\text{old}}(\tau_{i,t}|\tau_{i,<t})} \hat{A}_{i,t}, \right. \right.$$

$$\left. \left. \text{clip}\left( \frac{\pi_\theta(\tau_{i,t}|\tau_{i,<t})}{\pi_\theta^{\text{old}}(\tau_{i,t}|\tau_{i,<t})}, 1-\epsilon, 1+\epsilon \right) \hat{A}_{i,t} \right) - \beta \mathbb{D}_{KL} \left[ \pi_\theta || \pi_{\text{ref}} \right] \right] \quad (3)$$

$$\mathbb{D}_{KL} \left[ \pi_\theta || \pi_{\text{ref}} \right] = \frac{1}{\sum_{t=1}^{|\tau|} \mathbb{I}(\tau_t)} \sum_{t=1}^{|\tau|} \mathbb{I}(\tau_t) \cdot \left( \frac{\pi_{\text{ref}}(\tau_t|\tau_{<t})}{\pi_\theta(\tau_t|\tau_{<t})} - \log \frac{\pi_{\text{ref}}(\tau_t|\tau_{<t})}{\pi_\theta(\tau_t|\tau_{<t})} - 1 \right) \quad (4)$$

where $\epsilon$ and $\beta$ are set to 0.2 and 0.01 respectively, and the other hyperparameters are the same as those in DAPO training.

## I   BASELINE IMPLEMENTATION DETAILS

We evaluate M3-Agent against three types of baselines:

**Socratic Models** This baseline adapts the Socratic Models framework (Zeng et al., 2022), which uses a multimodal model to describe 30-second video clips. These descriptions are stored as long-term memory. To answer a question, an LLM performs retrieval augmented generation (RAG) (Lewis et al., 2020): It first invokes a `search_clip` function to retrieve memory relevant to the question, and then generates a response based on the retrieved content.

We implement both closed-source and open-source multimodal models for memory generation:

- Gemini-1.5-Pro (Team et al., 2024): Takes the full 30-second video clip as input.
- GPT-4o (Hurst et al., 2024): Since it does not process audio, we provide video frames sampled at 0.5 fps and ASR transcripts.
- Qwen2.5-Omni-7b (Xu et al., 2025a): An advanced open-source multimodal model that supports both visual and audio inputs. It receives the full video as input.

- Qwen2.5-VL-7b (Bai et al., 2025): An open-source vision-language models with SOTA results in visual-language tasks. Like GPT-4o, it receives both video frames (sampled at 0.5 fps) and ASR transcripts.

For all variants, GPT-4o serves as the LLM for RAG-based question answering. We perform extensive prompt engineering to optimize each setup. All prompts are listed in Appendix M.2.

**Online Video Understanding Methods** We further compare our approach with three online video understanding frameworks: MovieChat (Song et al., 2024), MA-LMM (He et al., 2024a), and Flash-VStream (Zhang et al., 2024b). Unless otherwise specified, we adopt their official pretrained weights and default configurations.

- MovieChat (Song et al., 2024): It uses a sliding-window to extract frame-level features and stores them in a hybrid memory; the LLM performs QA conditioned on this memory.

- MA-LMM (He et al., 2024a): It processes frames in an online manner, consisting of feature extraction (1 fps), temporal modeling (100-frame input), and LLM decoding.

- Flash-VStream (Zhang et al., 2024b): It adopts a two-stage asynchronous pipeline: stream video frame compression (1 fps), and LLM-based QA over the compressed features.

**Agent Methods** We also compare M3-Agent with agents implemented via prompting closed-source commercial models. Specifically, we consider the following two baselines:

- Gemini-Agent: Gemini-1.5-Pro is prompted separately for memory access and control process. During memory access, it receives the full video with audio, facial recognition results and speaker identification results to generate episodic and semantic memories, denoted as `memory-gemini-prompt`. In the control process, it performs memory searches and generates responses, referred to as `control-gemini-prompt`.

- Gemini-GPT4o-Hybrid: We also evaluate a setup where GPT-4o is prompted to perform memory search and generate responses (`control-gpt4o-prompt`). The memory access remains handled by `memory-gemini-prompt`.

The prompts are provided in Appendix M.3.

## J    CROSS-MODEL ASSESSMENT OF AUTOMATIC EVALUATION OF M3-BENCH

To ensure the robustness of results obtained by GPT-4o automatic evaluator, we perform a cross-model validation using two additional evaluators. Specifically, we employ Gemini-1.5-Pro and Qwen3-32B to evaluate all baselines and our M3-Agent on M3-Bench, using the same prompt as for GPT-4o shown in Table 23 . The corresponding results are presented in Table 15 and Table 16. Both alternative evaluators are consistent with the evaluations produced by GPT-4o shown in Table 4, thereby confirming the reliability of the automatic evaluation.

## K    STUDY ON MEMORY OF M3-AGENT

### K.1    ABLATION STUDY

We perform an ablation study to evaluate the conflict resolution mechanism in M3-Agent. Specifically, we remove the voting mechanism. When a conflict arises in memory, the system retains the most recent memory entry. The results, shown in Table 17, highlight the importance of the voting mechanism: without it, QA accuracy drops by 3.2% on M3-Bench-robot, 2.5% on M3-Bench-web, and 4.5% on Video-MME-Long. These declines demonstrate the voting mechanism is crucial for resolving contradictions and maintaining overall performance.

We ablate the contribution of different subtypes of semantic memory. Table 17 reports the results of removing attribute memory, relation memory, and rule memory. The results indicate that

Table 15: Results on M3-Bench-robot and M3-Bench-web, evaluated by Gemini-1.5-Pro automatic evaluator.

| Method | M3-Bench-robot | | | | | | M3-Bench-web | | | | | |
|---|---|---|---|---|---|---|---|---|---|---|---|---|
| | ME | MH | CM | PU | GK | **All** | ME | MH | CM | PU | GK | **All** |
| *Socratic Model* | | | | | | | | | | | | |
| Qwen2.5-Omni-7b | 2.2 | 1.7 | 1.7 | 1.6 | 2.1 | 2.2 | 9.1 | 9.2 | 14.4 | 10.9 | 14.4 | 11.6 |
| Qwen2.5-VL-7b | 3.4 | 3.5 | 3.6 | 4.6 | 4.0 | 3.8 | 11.5 | 10.3 | 13.2 | 14.2 | 21.7 | 15.2 |
| Gemini-1.5-pro | 6.9 | 7.8 | 7.6 | 9.9 | 8.6 | 8.3 | 18.5 | 18.4 | 23.3 | 23.1 | 30.0 | 23.7 |
| GPT-4o | 10.0 | 8.4 | 8.4 | 10.4 | 8.0 | 8.7 | 21.0 | 21.4 | 31.8 | 27.1 | 41.8 | 29.3 |
| *Online Video Understanding Methods* | | | | | | | | | | | | |
| MovieChat | 13.6 | 9.0 | 12.8 | 16.2 | 7.3 | 11.3 | 12.5 | 5.5 | 12.3 | 17.4 | 11.7 | 12.7 |
| MA-LMM | 26.5 | 24.3 | 24.4 | 39.2 | 17.1 | 25.5 | 27.4 | 10.7 | 22.9 | 39.9 | 17.4 | 25.1 |
| Flash-VStream | 22.8 | 20.9 | 20.1 | 26.1 | 15.3 | 20.5 | 24.6 | 10.9 | 25.1 | 32.1 | 20.7 | 23.9 |
| *Agent Method* | | | | | | | | | | | | |
| Gemini-Agent | 16.4 | 15.9 | 15.8 | 20.0 | 15.5 | 16.7 | 28.6 | 20.3 | 34.5 | 33.9 | 44.0 | 33.5 |
| Gemini-GPT4o-Hybrid | 21.6 | 25.0 | 22.2 | 28.4 | **22.5** | 23.7 | 36.5 | 27.0 | 37.3 | 44.5 | 52.8 | 41.7 |
| **M3-Agent** | **32.9** | **28.1** | **30.5** | **42.6** | 18.8 | **30.3** | **46.5** | **28.4** | **43.9** | **59.4** | **53.3** | **48.8** |

Table 16: Results on M3-Bench-robot and M3-Bench-web, evaluated by Qwen3-32B automatic evaluator.

| Method | M3-Bench-robot | | | | | | M3-Bench-web | | | | | |
|---|---|---|---|---|---|---|---|---|---|---|---|---|
| | ME | MH | CM | PU | GK | **All** | ME | MH | CM | PU | GK | **All** |
| *Socratic Model* | | | | | | | | | | | | |
| Qwen2.5-Omni-7b | 1.9 | 1.4 | 1.7 | 1.5 | 2.1 | 2.0 | 8.6 | 9.2 | 14.6 | 10.8 | 14.6 | 11.5 |
| Qwen2.5-VL-7b | 3.1 | 3.2 | 3.4 | 4.6 | 3.7 | 3.6 | 11.5 | 10.3 | 13.4 | 14.0 | 22.1 | 15.2 |
| Gemini-1.5-pro | 6.9 | 7.8 | 7.4 | 9.9 | 8.3 | 8.2 | 18.4 | 18.6 | 23.6 | 23.1 | 30.0 | 23.7 |
| GPT-4o | 9.6 | 8.4 | 8.2 | 10.4 | 8.0 | 8.5 | 21.1 | 21.7 | 32.5 | 27.1 | 42.0 | 29.5 |
| *Online Video Understanding Methods* | | | | | | | | | | | | |
| MovieChat | 13.4 | 8.7 | 13.2 | 15.9 | 8.0 | 11.3 | 12.7 | 5.5 | 12.0 | 17.4 | 12.0 | 12.8 |
| MA-LMM | 26.5 | 24.6 | 24.6 | 39.4 | 17.4 | 25.6 | 27.4 | 10.9 | 22.9 | 40.2 | 17.6 | 25.3 |
| Flash-VStream | 22.3 | 20.6 | 20.1 | 25.7 | 14.7 | 20.3 | 25.1 | 10.8 | 24.0 | 32.3 | 20.9 | 23.8 |
| *Agent Method* | | | | | | | | | | | | |
| Gemini-Agent | 16.1 | 17.3 | 15.8 | 20.2 | 15.1 | 16.9 | 29.7 | 19.5 | 33.6 | 33.8 | 44.0 | 33.2 |
| Gemini-GPT4o-Hybrid | 21.5 | 26.8 | 23.3 | 29.5 | **24.0** | 24.8 | 36.0 | 26.0 | 37.9 | 43.8 | 52.0 | 41.2 |
| **M3-Agent** | **31.6** | **27.8** | **29.6** | **42.3** | 17.7 | **29.6** | **45.1** | **28.4** | **42.8** | **59.7** | **53.9** | **48.8** |

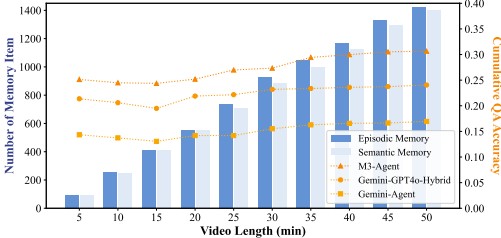

(a) Memory item growth and cumulative QA accuracy as video length increases.

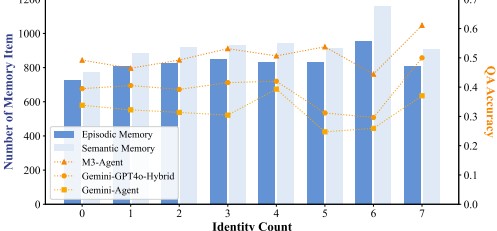

(b) Memory size and QA accuracy across varying numbers of identities.

Figure 5: Analysis of memory size and QA accuracy with increasing video duration and identity count.

Table 17: Ablation study of the conflict resolution mechanism and semantic memory types.

| Method | M3-Bench-robot | | | | | | M3-Bench-web | | | | | | Video-MME-Long |
|---|---|---|---|---|---|---|---|---|---|---|---|---|---|
| | ME | MH | CM | PU | GK | All | ME | MH | CM | PU | GK | All | |
| w/o voting | 28.6 | 26.6 | 26.1 | 39.1 | 18.7 | 27.5 | 41.8 | 20.1 | 40.1 | 57.8 | 51.9 | 46.4 | 57.3 |
| w/o attributes | 25.7 | 23.5 | 25.4 | 35.3 | 15.8 | 24.0 | 35.5 | 22.7 | 35.4 | 47.1 | 42.4 | 38.9 | 49.7 |
| w/o relations | 24.2 | 21.9 | 22.9 | 31.1 | 13.4 | 22.9 | 39.7 | 24.6 | 38.9 | 51.4 | 47.9 | 42.1 | 51.6 |
| w/o rules | 25.8 | 23.6 | 24.2 | 33.1 | 14.9 | 23.6 | 40.9 | 25.3 | 39.6 | 53.5 | 49.0 | 42.7 | 54.9 |
| **M3-Agent** | 32.8 | 29.4 | 31.2 | 43.3 | 19.1 | 30.7 | 45.9 | 28.4 | 44.3 | 59.3 | 53.9 | 48.9 | 61.8 |

## K.2 PERFORMANCE SCALING WITH VIDEO DURATION

To evaluate how memory growth affects the memory system, we plot memory size and cumulative QA accuracy on M3-Bench-robot as functions of video duration, as shown in Figure 5a. The results show that memory size increases approximately linearly with video length. QA accuracy increases slightly as videos become longer, indicating that the system maintains robust performance even as memory grows. Furthermore, M3-Agent consistently outperforms the strongest baselines, prompting-based agents using Gemini-1.5-Pro and GPT-4o across all video durations.

## K.3 IMPACT OF IDENTITY GROWTH

To examine how identity count affects the memory system, we plot memory size and QA accuracy against on the number of identities on M3-Bench-web. We use M3-Bench-web because it provides a wide range of identity counts while keeping video length approximately constant (an essential control, as video length is the dominant factor affecting memory size). The results are shown in Figure 5b. we find that both memory size and QA accuracy remain largely stable as the number of identities increases. Notably, M3-Agent consistently surpasses the strongest baselines across all identity-count categories. For videos with many identities, the performance gap widens more, demonstrating that M3-Agent is more robust than baseline methods when handling more complex scenarios.

## L CASE STUDY

**Entity-Centric Long-Term Memory vs. Standard Memory** Table 18 compares M3-Agent memory with traditional memory generated by Gemini-1.5-Pro and GPT-4o. M3-Agent features greater consistency due to its entity-centric memory structure and accumulates richer world knowledge through semantic memory generation.

M3-Agent maintains consistent person identities. Even when speakers' faces never appear on screen, M3-Agent correctly distinguish them and associate their voices with `<character_0>` or `<character_16>` in long-term memory. In contrast, Gemini-1.5-Pro describes them only as "an off-screen voice", while GPT-4o fails to assign any consistent identity.

M3-Agent generates semantic memory in addition to episodic memory, effectively expanding the utility of the memory. For example, answering "Is Lucas skilled as cooking?" can not rely on episodic memory alone, as relevant events cannot be retrieved by key words such as Lucas or cooking skills. However, the semantic memory "`<character_23>` is named Lucas; `<character_23>` demonstrates a willingness to learn and improve cooking skills" is retrievable and directly supports correct answer.

**Memorization** Table 20, 21 present two examples illustrating the episodic and semantic memories generated during memory access. Compared to `memory-gemini-prompt`, `memory-7b-sft` demonstrates (1) more detailed episodic memory generation, including richer scene descriptions, character actions and expressions, and dialogue; (2) improved recognition of identity equivalence, enabling consistent long-term tacking of human identities; and (3) richer semantic memory extraction, proactively generating knowledge about characters and environments.

---

**Video:** https://www.youtube.com/watch?v=6Unxpxy-Ct4     **Time:** 14:00 - 14:30 (also shown in Figure 2)
**Question:** Is Lucas skilled at cooking?

---

**Gemini-1.5-Pro:**

In a brightly lit kitchen, Lucas, wearing a dark short-sleeved shirt and dark pants, stands at a marble countertop, carefully chopping green onions on a wooden cutting board.

A voice off-screen, instructs him: "Use the best part of the knife and let the knife cut. Very nice, easy."

Another voice, also off-screen, exclaims, "This is really hard!"

Lucas's father replies, "The flat side should be on the bottom, too."

Lucas then picks up the sliced onion greens from the board, leaving the white parts.

Lucas's father asks, "Alright, tell me that it looks..." Lucas's hand is blurred, but he appears to be brushing away the cut onion pieces.

The off-screen voice then says, "Okay, this is Lucas's second attempt." Lucas slices some of the white parts of the onion into sections.

He stops to put on a pair of glasses, saying "Oh, I can see." Now wearing glasses, Lucas resumes chopping the white parts of the onions.

The other off-screen voice comments, "Oh, that's pretty good!"

Lucas's father advises, "Your angle on your knife is a little too high. Keep it low."

Lucas adjusts his knife angle, as his father says, "Yeah, perfect. Good job."

Lucas continues chopping the remaining white parts of the onions.

---

**GPT-4o:**

The video clip captures a casual scene in a modern kitchen, highlighting a learning moment between two individuals. The setting features a spacious kitchen with clean, white cabinetry, glass-paneled cupboards showcasing neatly stacked dishes and bowls, marble countertops, and stainless steel appliances, including a stove with distinct red knobs. The scene appears well-lit, with warm and even lighting creating a welcoming atmosphere suitable for instructional interaction.

In the first segment of the clip, one individual, wearing a plain white long-sleeve shirt, leans over the counter on one side, providing guidance to the other person, who is actively chopping ingredients. This individual speaks calmly, offering encouraging and instructional remarks: "Use the best part of the knife and let the knife cut. Very nice and easy." Their tone is steady, suggesting patience and mentoring. They exhibit relaxed body language, with arms resting on the counter, adding to the informal and supportive dynamic.

The other individual, wearing a black t-shirt and a silver chain bracelet on their left wrist, stands on the opposite side of the counter. They are focused on slicing long green onions (scallions) into thin julienne cuts on a wooden cutting board. Their movements are deliberate, but not entirely confident, as they appear to be learning or refining their knife skills. At one point, they express mild frustration, stating, "This is really hard," while continuing to adjust their cutting technique. As this occurs, the encouraging individual responds affirmatively with short phrases such as "Yeah" to acknowledge their effort.

In a noticeable shift, the person chopping announces, "Okay, this is Lucas's second attempt," referring to themselves in third person, emphasizing their ongoing learning process. They temporarily pause their activity to put on glasses, stating, "Wait. I'm going to put on my glasses this time," followed by an exclamation, "Oh, I can see," suggesting improved confidence or clarity in their ability to focus on the task at hand. This moment draws a lighthearted reaction, subtly changing the atmosphere to one of optimism.

The individual in the white shirt continues to provide constructive feedback, guiding Lucas on their knife angle, saying, "Your angle on your knife is a little too high." As Lucas adjusts their technique, the mentor reassures them: "Keep it low. Yeah, perfect. Good job." This results in tangible improvement, as Lucas successfully slices the scallions more precisely, creating a uniform pile of thin pieces on the cutting board.

Throughout the clip, the interaction conveys a sense of camaraderie and support in a calm, domestic setting. The kitchen environment, combined with the dialogue and actions, suggests an emphasis on skill-sharing and guidance, perhaps during a light evening activity. The modern, tidy surroundings reflect an atmosphere conducive to bonding and learning cooking techniques.

---

**M3-Agent:**

Episodic

<character_0> instructs <character_23> on how to slice scallions, emphasizing using the best part of the knife and letting the knife do the work.

<character_0> comments that slicing is not easy.

<character_16> says, "This is really hard!"

<character_23> wearing a black shirt slices scallions on a wooden cutting board placed on a white marble countertop.

<character_0> tells the person to place the flat side of the scallion on the bottom.

<character_16> says, "Okay, this is Lucas's second attempt."

<character_23> attempts to put on his glasses.

<character_23> says, "Oh, I can see."

<character_23> continues slicing the scallions.

<character_16> comments, "Oh, that's pretty good!"

<character_0> comments that the person's knife angle is a little too high.

<character_0> instructs the person to keep the knife low and praises their slicing as perfect and a good job.

<character_23> continues slicing the scallions with the knife held lower.

The kitchen has white cabinets with glass doors, a white subway tile backsplash, and stainless steel appliances.

There are plates and glasses visible inside the cabinets.

Semantic

<character_0> displays patience and a supportive attitude.

<character_23> is named Lucas.

<character_23> demonstrates a willingness to learn and improve culinary skills.

<character_0> takes on an instructive role, offering specific feedback to <character_23>.

<character_0> is the father of <character_23>.

The video demonstrates a culinary lesson focused on the proper technique for slicing scallions.

The overall tone is encouraging and supportive, fostering a positive learning environment.

The feedback provided contributes to a clear cause-and-effect relationship between knife handling technique and the quality of the slice.

The video likely takes place in a home kitchen, given the casual atmosphere and presence of everyday kitchenware.

The act of slicing scallions is a common culinary skill, often practiced in various cuisines.

The wooden cutting board serves as a safe and stable surface for chopping, preventing damage to knives and countertops.

The instruction on using the 'best part' of the knife and keeping the flat side on the bottom are common knife skills for efficient and safe chopping.

The use of glasses by <character_23> demonstrates the importance of vision in precise tasks like cooking.

The stainless steel appliances and white subway tile backsplash are common features of modern kitchen design.

---

Table 18: Comparison of M3-Agent memory with traditional memory generated by Gemini-1.5-Pro and GPT-4o. M3-Agent features greater consistency due to its entity-centric memory structure and accumulates richer world knowledge through semantic memory generation.

**Control** To illustrate the control process in detail, Table 22 presents a complete generation trajectory of `control-32b-rl`. The input question is: "Is Tomasz a person with rich imagination or someone who lacks imagination?"

In the first round, the agent searches its memory for Tomasz's character ID. In the second round, having identified Tomasz as `<character_4>`, it attempts a direct query: "What is `<character_4>`'s personality regarding imagination?" Finding no relevant memory in the third round, the agent reasons based on `<character_4>`'s role as CTO of a company and generates a more targeted query: "What are `<character_4>`'s creative problem-solving methods?" This yields a relevant memory: "`<character_4>` is innovative and forward-thinking, as evidenced by his interest in scaling drone technology for personal flight."—a piece of semantic memory. By the fourth round, the agent has collected enough information in its context to generate the final answer.

**Hard Case in M3-Bench** The accuracy of various methods demonstrates that M3-Bench, particularly M3-Bench-robot, presents a significant challenge. We perform a detailed error analysis of M3-Agent on M3-Bench. We randomly sample 50 failure cases from M3-Agent's responses on M3-Bench and examine their underlying causes. The resulting error categories are summarized in Table 19.

Table 19: Summary of common error types.

| Error Type | Percentage | Explanation |
|---|---|---|
| Reasoning about fine-grained details | 50% | The agent fails to extract precise, detailed information from its observations. |
| Spatial reasoning | 20% | The agent fails to understand spatial layout or tracking spatial changes. |
| Incorrect search strategy | 12% | The agent does not correctly decompose the query or fails to execute an effective search plan to gather necessary information. |
| Missing Human names | 10% | The agent fails to extract the relevant character names required to answer the question. |
| Others | 8% | |

The most common error type involves reasoning about fine-grained details. For instance, questions like "Who wants to eat the ham sausage?" or "Which coat rack should Emma's hat be laced, taller one or shorter one?" require the agent to extract precise information from its observations. However, retaining all such details in memory is impractical and may cause cognitive overload. To address this, the agent must use attention mechanisms that enables selective memorization. During execution, it can develop task-specific world knowledge, allowing it to focus on relevant details while ignoring the irrelevant, thereby improving task performance.

Another category of hard cases is related to spatial reasoning. In the M3-Bench-robot, a number of questions challenge the agent's capability on spatial cognition, such as understanding spatial layout and tracking spatial changes. Examples include: "Where can the robot get the snacks?" and "Is Leo's water cup currently on the second or third shelf from the top of the rack?" Since verbal memory is generally less effective than visual memory for retaining spatial information, the long-term memory should be designed to incorporate richer visual content, e.g., snapshots, to better support spatial reasoning.

`Video` (Illustrated as 12 frames)

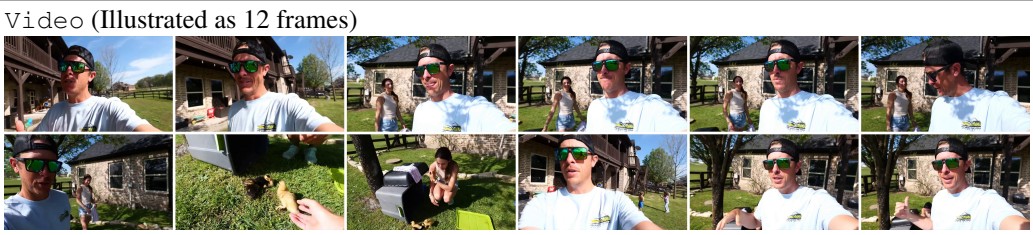

Faces

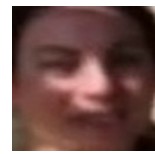 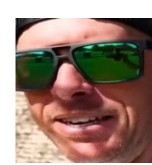

<face_1413>     <face_1412>

Dialogues
• <voice_1>: (00:03-00:12) Every, but I feel like every sibling dynamic is different. I was the youngest and I just took it. I just took it. I didn't have the aggression to fight back like
• <voice_2>: (00:00-00:02) these experiences through my children, (00:11-00:16) close to, uh, it's neck and neck. Yeah, they they go after each other. (00:24-00:29) we're gonna have the fish coming to the fish feed and then we're gonna have ducks coming too. It's gonna be a
• <voice_812>: (00:17-00:23) Wow. So we're gonna we're gonna go fishing at the pond. We're gonna I'm gonna get a fish feeder and we're gonna have

**memory-gemini-prompt**

Episodic memory
• <face_1412> is wearing a light blue t-shirt with a graphic design, a black baseball cap, and mirrored sunglasses.
• He is standing in a backyard with a brick house, a wooden deck, a fence, and some trees.
• <voice_2> says he's reliving experiences through his children.
• A woman, presumably his wife, walks by in the background carrying some items.
• <voice_1> says every sibling dynamic is different and that she was the youngest and just took it.
• <voice_2> says the children are neck and neck and go after each other.
• <face_1412> and the woman are holding small yellow ducklings.
• <voice_812> says they're going fishing at the pond and will get a fish feeder.
• Two children are playing in the backyard with squirt guns.
• <voice_812> says they'll have fish coming to the fish feed and ducks coming too.

Semantic memory
• <face_1412> appears to be a father who enjoys spending time with his family and doing outdoor activities.
• <face_1412> and <face_1413> have a casual and comfortable relationship.
• The video depicts a family preparing for a fishing outing at their pond and discussing their children's sibling dynamics.
• The setting is a rural or suburban home with a large backyard and a pond.

**memory-7b-sft (Ours)**

Episodic memory
• <face_1412> wears a white t-shirt with a green and yellow graphic design, a black baseball cap, and green mirrored sunglasses.
• <face_1412> is outdoors on a sunny day, in a grassy area with a brick building and a wooden fence in the background.
• <face_1412> speaks to the camera, saying, "these experiences through my children, which is"
• <face_1413> joins <face_1412>, wearing a white tank top and denim shorts.
• <voice_1> says, "Every, but I feel like every sibling dynamic is different. I was the youngest and I just took it. I just took it. I didn't have the aggression to fight back like"
• <face_1412> points a finger upwards.
• <voice_2> says, "close to, uh, it's neck and neck. Yeah, they they go after each other."
• <face_1413> and <face_1412> hold two small, fluffy yellow ducklings.
• The ducklings are placed on the grass near a black pet carrier.
• <face_1413> kneels by the pet carrier, holding a yellow duckling.
• <voice_812> says, "Wow. So we're gonna we're gonna go fishing at the pond. We're gonna I'm gonna get a fish feeder and we're gonna have"

*(Continued on next page)*

- `<face_1412>` gestures with his hand.
- `<voice_2>` says, "we're gonna have the fish coming to the fish feed and then we're gonna have ducks coming too. It's gonna be a"
- A child in a blue outfit is visible in the background.

```
Semantic memory
```
- Equivalence: `<face_1412>`, `<voice_812>`
- Equivalence: `<face_1412>`, `<voice_2>`
- Equivalence: `<face_1413>`, `<voice_1>`
- `<face_1412>` exhibits a relaxed and informal demeanor.
- `<face_1412>` displays an interest in outdoor activities and family-oriented experiences.
- `<face_1413>` demonstrates a nurturing and caring personality.
- `<voice_1>` appears to be reflective and analytical when discussing personal experiences.
- `<voice_2>` expresses enthusiasm and anticipation for future activities.
- `<voice_812>` seems knowledgeable about pet care and enjoys engaging in outdoor recreational activities.
- `<face_1412>` and `<face_1413>` appear to have a comfortable and supportive relationship, likely romantic partners or close family members.
- `<voice_1>` and `<face_1412>` share a close bond, possibly as parent and child, marked by open communication and shared experiences.
- The video depicts a slice-of-life scene centered around family interaction and leisure activities.
- The narrative progresses from a reflective discussion about sibling relationships to a forward-looking anticipation of shared activities.
- The overall tone is lighthearted and positive, emphasizing the joys of family connection and shared experiences.
- The video showcases a blend of indoor and outdoor settings, creating a dynamic visual narrative.
- The video likely belongs to the family vlog genre, documenting everyday life and experiences.
- The video showcases a typical suburban or rural setting with a house, fence, and outdoor space.
- Keeping ducklings requires specific pet care knowledge, including appropriate housing and feeding.
- The video implies the cultural norm of family members engaging in shared recreational activities.
- Fishing is a popular outdoor leisure activity often enjoyed in natural settings like ponds.
- The video suggests the practice of using a fish feeder to attract and manage fish populations in a pond.

Table 20: Comparison of memory generations between `memory-7b-sft` and `memory-gemini-prompt` for **case KHslnSzK2SU, clip 23:00-23:30, M3-Bench-web**. Yellow highlights indicate additional important details provided by our model.

---

`Video` (Illustrated as 12 frames)

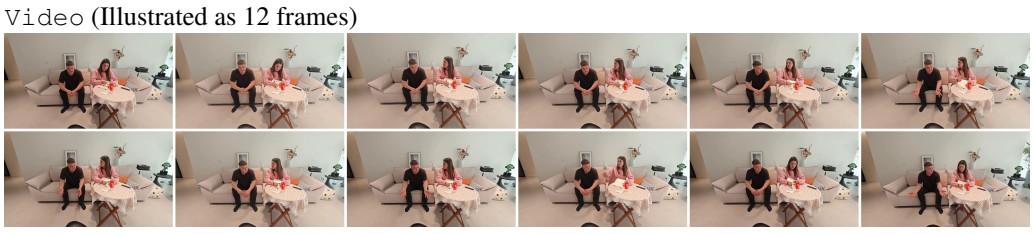

`Faces`

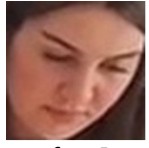 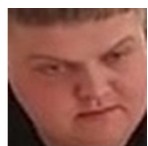

<face_5>   <face_4>

Dialogues

• <voice_53>: (00:00-00:03) Robot, can you at least bring some fun games or puzzles we can play? (00:13-00:17) What am I supposed to do? Sometimes they come with no pieces. What am I supposed to do? (00:20-00:22) Well, what am I supposed to? I'm bored. I want to play a game.

• <voice_204>: (00:04-00:08) Really? Puzzle is wasting time for you. You don't like puzzles. (00:08-00:13) But every time you play puzzle, you miss many pieces of it, which I love. (00:23-00:27) I don't know. Go out and play some games, some games outside.

• <voice_316>: (00:18-00:20) So just leave it. Just don't take it.

### memory-gemini-prompt

Episodic memory

• <face_4> sits on a light-colored couch, facing <face_5>, who sits at a small table.

• <face_5> wears a light pink sweatshirt, <face_4> wears a dark shirt and dark pants.

• <voice_53> asks <face_5> to bring some fun games or puzzles.

• <face_5> tells <voice_53> that puzzles are a waste of time.

• <face_5> explains that <voice_53> always loses puzzle pieces.

• <voice_53> expresses frustration about missing puzzle pieces and asks what to do.

• <face_5> suggests <voice_53> just leave the puzzle incomplete.

• <voice_53> complains of boredom and wanting to play a game.

• <face_5> suggests <voice_53> go outside and play.

• <voice_53> asks if <face_5> dislikes Legos too, because Lego pieces can also get lost.

Semantic memory

• <face_4>/<voice_53> appears bored and frustrated.

• <face_5>/<voice_204> appears slightly annoyed or dismissive.

• <face_4>/<voice_53> and <face_5>/<voice_204> seem to have a casual, possibly familial relationship.

• The interaction revolves around <face_4>/<voice_53>'s boredom and <face_5>/<voice_204>'s suggestions for alternative activities.

• The setting appears to be a living room or family room.

### memory-7b-sft (Ours)

Episodic memory

• <face_4> sits on a beige couch, wearing a black t-shirt and black pants.

• <face_5> sits next to <face_4> on the couch, wearing a pink sweatshirt.

• A small round table with a red and white checkered tablecloth is placed between <face_4> and <face_5>.

• On the table are a bottle of red liquid, a white cup, a notebook, and a pen.

• A vase with dried flowers and a decorative metal piece are on a small table behind the couch.

• A projector and a small green figurine are visible in the background.

• <voice_53> asks the robot to bring games or puzzles.

• <voice_204> tells <voice_53> that puzzles are a waste of time and that <voice_53> doesn't like puzzles.

• <voice_204> says that <voice_53> always misses pieces when doing puzzles, which <voice_204> enjoys finding.

• <voice_53> expresses frustration, saying that sometimes puzzles come with no pieces.

• <voice_204> suggests that <voice_53> just leave the puzzle if they don't want to do it.

• <voice_53> says they are bored and want to play a game.

• <voice_204> suggests that <voice_53> go outside and play games.

• <voice_53> asks if <voice_204> doesn't like Legos either, because sometimes Lego pieces are missing.

- `<voice_204>` responds with "I don't know".
- `<face_5>` gestures with their hands while speaking.
- `<face_4>` gestures with their hands while speaking.
- `<face_5>` writes in the notebook while speaking.

`Semantic memory`

- Equivalence: `<face_4>`, `<voice_53>`
- Equivalence: `<face_4>`, `<voice_316>`
- Equivalence: `<face_5>`, `<voice_204>`
- `<voice_53>` appears to be easily bored and seeks immediate entertainment.
- `<voice_204>` displays a pragmatic and solution-oriented approach to problem-solving.
- `<voice_204>` seems to enjoy challenges and finding solutions, even in seemingly mundane tasks like puzzles.
- `<voice_53>` expresses a preference for active recreation over quiet, puzzle-based activities.
- `<voice_204>` and `<voice_53>` appear to have a comfortable, familiar dynamic, likely indicative of a close relationship.
- The interaction between `<voice_204>` and `<voice_53>` suggests a playful banter, with `<voice_204>` teasing `<voice_53>` about their attitude.
- `<voice_204>` takes on a mentoring or guiding role, offering suggestions and solutions to `<voice_53>`'s requests.
- The scene depicts a common household scenario of differing preferences and mild conflict regarding leisure activities.
- The narrative unfolds as a request for entertainment, followed by disagreement on suitable options and a suggestion for a more intellectually engaging activity.
- The overall tone is casual and familiar, reflecting the dynamics of a comfortable, possibly familial or romantic, relationship.
- The scene showcases a power dynamic where one individual attempts to influence the other's choices, leading to a mild power negotiation.
- The video likely takes place in a private residence, possibly an apartment or a recently built home, given the modern interior design.
- The presence of a projector suggests a focus on entertainment and media consumption within the household.
- The availability of puzzles and Lego sets implies a focus on recreational activities and potentially family-oriented entertainment.
- The red liquid in the bottle is likely a type of soda or juice, commonly consumed as a refreshing beverage during leisure time.
- The checkered tablecloth on the table signifies a casual dining or activity setup, possibly for a snack or a casual game session.
- The dried flowers and decorative metal piece function as home decor, contributing to the overall aesthetic and ambiance of the space.
- The small green figurine may be a collectible item or a decorative ornament, adding a personal touch to the room's decor.

Table 21: Comparison of memory generations between `memory-7b-sft` and `memory-gemini-prompt` for **living_room_22, clip 4:00-4:30, M3-Bench-robot**. Yellow highlights indicate additional important details provided by our model.

---

**Question:** Is Tomasz a person with rich imagination or someone who lacks imagination?
**Answer:** A person with rich imagination.

**system** `(system_prompt)`

*(Continued on next page)*

You are given a question and some relevant knowledge. Your task is to reason about whether the provided knowledge is sufficient to answer the question. If it is sufficient, output [Answer] followed by the answer. If it is not sufficient, output [Search] and generate a query that will be encoded into embeddings for a vector similarity search. The query will help retrieve additional information from a memory bank.

Question: What is Theresa Yeap's attitude like in her explanation of emergency services?

---

**user** (instruction_prompt)

Searched knowledge: {}

Output the answer in the format:

Action: [Answer] or [Search]

Content: {content}

If the answer cannot be derived yet, the {content} should be a single search query that would help retrieve the missing information. The search {content} needs to be different from the previous.

You can get the mapping relationship between character ID and name by using search query such as: "What is the name of <character_{i}>" or "What is the character id of {name}".

After obtaining the mapping, it is best to use character ID instead of name for searching.

If the answer can be derived from the provided knowledge, the {content} is the specific answer to the question. Only name can appear in the answer, not character ID like <character_{i}>.

---

**assistant**

<think> Okay, let's see. The user is asking whether Tomasz is a person with rich imagination or someone who lacks imagination. The problem is that there's no provided knowledge about Tomasz in the given context. The Searched knowledge is empty, so I don't have any information to base an answer on.

First, I need to check if the existing knowledge can answer the question. Since the knowledge is empty, there's nothing to reference. The user mentioned that if the answer can't be derived, I should output a search query. The instructions also say to use character IDs if possible, but since we don't know Tomasz's ID yet, I need to find that first.

So the first step is to find the character ID for Tomasz. The user suggested using a query like "What is the character id of Tomasz" to get that mapping. Once I have the ID, I can then search for information related to his imagination. But without the ID, any search might not be effective. Therefore, the initial search should be to find the character ID. Then, using that ID, look into his traits. Since the current knowledge is empty, the correct action is to search for the character ID first.</think>

Action: [Search]

Content: What is the character id of Tomasz

---

**user**

Searched knowledge: {"CLIP_4": ["<character_4>'s name is Tomasz Patan.", "The man in the white shirt, now identified as Tomasz Patan, Founder & Chief Technology Officer of Jetson, speaks to the camera in a workshop setting."], "CLIP_5": ["<character_4>'s name is Patán.", "<character_2> identifies <character_4> as Patán, the founder and CTO of Jetson, a company developing an eVTOL vehicle."], "CLIP_8": ["<character_2>'s name is Angus Whitley."], "CLIP_15": ["<character_0>'s name is Akshat Rathi.", "<character_0> wears glasses and has a beard."], "CLIP_17": ["<character_5>'s name is Matthew Clark."], "CLIP_20": ["<character_3>'s name is Tom Muniz.", "<character_3> identifies himself as Tom Muniz, Chief Technology Officer of Archer Aviation."], "CLIP_21": ["<character_7> is likely a pilot or flight controller, given their interaction with air traffic control."], "CLIP_26": ["<character_2>'s name is Robin Riedel.", "<character_2>, wearing glasses and a light blue shirt, is identified as Robin Riedel, a partner at McKinsey & Company and co-leader of the McKinsey Center for Future Mobility.", "<character_2> is a certified commercial pilot."], "CLIP_31": ["The graphic then shows a user profile icon appearing, representing 'pilot requirements'."], "CLIP_36": ["<character_2> is shown again in the library."], "CLIP_37": ["<character_1> displays a cautious and rule-abiding personality."], "CLIP_40": ["<character_12>'s name is Isabel Metz.", "<character_12> identifies the Hamburg airport on the map."], "CLIP_42": ["A name tag identifies <character_8> as 'Bianca Schuchardt, Researcher, German Aerospace Center'."]}

---

---

instruction prompt

---

**assistant**

<think> Okay, let's see. The user wants to know if Tomasz has a rich imagination or lacks it. From the previous search, we found that Tomasz Patan is <character_4>. Now, I need to check the provided knowledge to see if there's any info about his imagination.

Looking through the CLIP entries for <character_4> (Tomasz Patan), the info given is about his role as Founder & CTO of Jetson, speaking in a workshop. There's no mention of his imagination. Other clips mention other characters but not their traits related to imagination. Since there's no data here about Tomasz's imagination, I can't derive an answer. The next step is to search for information on his imagination using his character ID. The query should be about his imagination, so maybe "Does ¡character_4> have rich imagination?" or similar. Need to make sure the query is different from previous ones. Let's go with "What is <character_4>'s personality regarding imagination?" to retrieve relevant info.</think>

Action: [Search]

Content: What is <character_4>'s personality regarding imagination?

**user**

Searched knowledge: {}

(The search result is empty. Please try searching from another perspective.)

instruction prompt

**assistant**

<think> Okay, let's see. The user is asking if Tomasz is a person with rich imagination or someone who lacks imagination. So far, we've tried searching and couldn't find any info about his imagination. The first step was to get his character ID, which is <character_4>. Then, we tried searching for his personality regarding imagination, but that came back empty.

---

Since there's no direct information about Tomasz's imagination in the provided knowledge, I can't derive an answer from the existing data. The user's instructions say that if the answer can't be derived yet, I should generate a new search query. The previous query was about his personality regarding imagination, which didn't yield results. So I need to approach it from another angle.

Maybe instead of asking about his personality, I can look for any mentions of his traits in general. Perhaps there's a clip where his characteristics are discussed, but it's not related to imagination. Alternatively, maybe his role as a CTO or founder of a company could imply something about his creativity or innovation, which are related to imagination. However, the knowledge provided doesn't mention anything about his traits beyond his professional role.

Given that the knowledge is still empty, I should create a new query that's different from the previous one. Perhaps asking about his creative abilities or how others perceive his problem-solving skills. Alternatively, checking for any quotes or statements he made that might indicate his imaginative nature.

The key is to generate a query that's specific enough to retrieve information if it exists. So maybe "What are <character_4>'s creative problem-solving methods?" or "How does <character_4> approach innovation in his work?" These queries might uncover information about his imagination through his work style or achievements.

Since the user wants the search query to be different from the previous ones, I'll go with one of these alternatives. Let's pick "What are <character_4>'s creative problem-solving methods?" to see if that yields any relevant information about his imagination.</think>

Action: [Search]

Content: What are <character_4>'s creative problem-solving methods?

**user**

---

Searched knowledge: {"CLIP_6": ["<character_4>, wearing a white polo shirt with the 'Jetson' logo, stands in a room with a partially assembled aircraft frame behind him.", "<character_4> holds a clear plastic cup and gestures with his hands as he speaks.", "<character_4> says, 'test and we have lots of data from it that we had redundant sensors on the vehicle that day.' ", "A person wearing a yellow vest labeled 'TEST TEAM' adjusts a red propeller on the aircraft outdoors.", "Another person wearing a black jacket and white and orange sneakers steps onto a scale.", "The person in the black jacket, now wearing a Sparco racing suit and helmet, walks towards an open field.", "<character_4> says, 'So for 11 years, I've been doing drones with camera gimbals.' ", "<character_4> continues, 'These drones were getting bigger and bigger.' ", "<character_4> says, 'And at some point, I asked myself the question, what if we scaled it?' ", "<character_4> puts on a Sparco helmet and fastens the chin strap.", "<character_4> adjusts the visor on his helmet.", "<character_4> buckles the seatbelt inside the aircraft.", "<character_4> says, 'So I could sit in there and pilot myself.' ", "<character_4> sits in the aircraft as it takes off, raising one hand in a gesture.", "The aircraft is white and black with four propellers.", "<character_4> is an engineer or designer involved in the development of the Jetson aircraft.", "<character_4> has a background in working with drones and camera gimbals, demonstrating expertise in aerial technology.", "<character_4> is innovative and forward-thinking, as evidenced by his interest in scaling drone technology for personal flight.", "<character_4> possesses a hands-on, experimental approach to his work, as shown by his involvement in testing and data analysis.", "<character_4> displays a sense of pride and enthusiasm for his work on the Jetson project.", "The video follows a narrative arc of innovation and realization, starting with the concept of scaling drone technology and culminating in a successful test flight.", "The video showcases a blend of technical detail and personal narrative, offering insights into both the engineering process and the driving force behind the project.", "The overall tone of the video is optimistic and forward-looking, emphasizing the potential of personal aerial vehicles.", "The video implicitly highlights the importance of data analysis and redundant sensor systems in ensuring flight safety and reliability.", "The video likely belongs to the genre of technology demonstration or promotional content, showcasing the Jetson aircraft.", "The setting includes both indoor and outdoor locations, suggesting a combination of design, testing, and operational phases of the project.", "The use of safety gear like helmets and vests indicates adherence to safety protocols in aviation and engineering environments.", "The presence of a test team suggests a structured approach to quality control and validation of the aircraft's performance.", "The scale model aircraft implies a phased approach to development, starting with smaller prototypes before moving to full-sized vehicles.", "The video showcases procedural norms in aviation, such as pre-flight checks, helmet usage, and controlled testing environments.", "The Jetson aircraft represents a novel application of drone technology, aiming to bridge the gap between unmanned aerial vehicles and personal flight.", "The video implies a growing trend of personal aerial mobility, driven by technological advancements and innovative design.", "The video's narrative structure follows a common format in technology demonstrations, starting with the problem statement (drones' size limitations) and presenting a solution (the Jetson aircraft).", "The video's focus on data analysis suggests a data-driven approach to design and optimization in the development of the Jetson aircraft."]}
`instruction_prompt`

**assistant**
<think> Okay, let's tackle this question. The user wants to know if Tomasz is a person with rich imagination or someone who lacks imagination. From the provided knowledge, we have a bunch of clips about Tomasz Patan (character_4). Let me go through the info again.
In CLIP_6, there's a lot of detail about his work with drones and the Jetson aircraft. He mentions scaling drone technology for personal flight, which shows he's thinking beyond the current applications. The notes say he's innovative and forward-thinking, with a hands-on approach. He's involved in testing and data analysis, and he's enthusiastic about his work. The video's tone is optimistic, highlighting his problem-solving methods and the development process.

*(Continued on next page)*

The key points here are his background in drones, his idea to scale them into personal aircraft, and his active role in testing. The description explicitly states he's innovative and forward-thinking. These traits are closely related to having a rich imagination. Since he's taking existing technology (drones) and envisioning a new application (personal flight), that requires creativity and imagination. Also, the narrative of the video presents him as someone who asks "what if" questions, which is a hallmark of imaginative thinking.
I don't see any information suggesting he lacks imagination. All the provided details support the idea that he's imaginative. Therefore, the answer should be that Tomasz is a person with rich imagination.</think>
Action: [Answer]
Content: Tomasz is a person with rich imagination.

Table 22: A complete generation trajectory in the control process.

## M  PROMPT TEMPLATES

### M.1  PROMPT FOR AUTOMATIC EVALUATOR OF M3-BENCH

Table 23 presents the prompt used by GPT-4o to assess M3-Bench.

---

**The prompt for GPT-4o evaluation**

You are provided with a question, a ground truth answer, and an answer from an agent model. Your task is to determine whether the ground truth answer can be logically inferred from the agent's answer, in the context of the question.

Do not directly compare the surface forms of the agent answer and the ground truth answer. Instead, assess whether the meaning expressed by the agent answer supports or implies the ground truth answer. If the ground truth can be reasonably derived from the agent answer, return "Yes". If it cannot, return "No".

Important notes:
• Do not require exact wording or matching structure.
• Semantic inference is sufficient, as long as the agent answer entails or implies the meaning of the ground truth answer, given the question.
• Only return "Yes" or "No", with no additional explanation or formatting.

Input fields:
• question: the question asked
• ground_truth_answer: the correct answer
• agent_answer: the model's answer to be evaluated

Now evaluate the following input:

Input:
• question: {question}
• ground_truth_answer: {ground_truth_answer}
• agent_answer: {agent_answer}

Output ('Yes' or 'No'):

---

Table 23: Prompt used by GPT-4o to evaluate M3-Bench.

### M.2  PROMPTS FOR SOCRATIC MODELS

Table 24 presents the prompt used in Socratic Models baselines. Through prompt engineering, we find that placing the question after the long context (e.g., video detailed descriptions) enhances the

model's ability to retain the question and focus on relevant information, leading to improved answer accuracy. Accordingly, in our Socratic Models experiments, we adopt this approach by appending the question to the end of the retrieved clip descriptions during the RAG-based QA stage.

---

**Caption Generation Prompt (Gemini-1.5-Pro, Qwen-2.5-Omni)**

---

You are an advanced video description generator tasked with providing a detailed, cohesive description of a video clip.
Follow these high-level principles to ensure your output is accurate and meaningful:
1. Focus on Observable Content.
2. Provide Context for the Environment and Timing.
3. Incorporate Audio Dialogue Information.
You are provided with a current video clip. (**GPT-4o, Qwen2.5-VL-7b Variant:** *You are provided with 15 key frames from a current video clip and audio text information ¡a list where each item represents a speech segment dict with the following fields: start time, end time, asr. The time information is the time in the current clip and not the global time¿.*)

Your Task:

Based on the video clip, generate a detailed and cohesive description of the video clip. The description should focus on the entire event, incorporating all relevant aspects of the characters, their actions, spoken dialogue, and interactions in a narrative format. The description should include (but is not limited to) the following categories:

1. Characters' Appearance: Describe the characters' appearance, including their clothing, facial features, body language, or any distinguishing characteristics that are noticeable in the frames.
2. Characters' Actions & Movements: Describe specific gestures, movements, or interactions performed by the characters. Include both major and minor actions that contribute to the overall scene, emphasizing any transitions between different actions.
3. Characters' Spoken Dialogue: Use the provided audio dialogue information to accurately transcribe or summarize the dialogue spoken by the characters. Include emotional tone, volume, or context if relevant (e.g., shouting, whispering, laughing).
4. Characters' Contextual Behavior and Attributes: Describe the characters' roles in the scene, their emotional states, motivations, or relationships with other characters. Highlight any conflict, bonding, or change in dynamics.
5. Environmental Context: Include relevant details about the environment where the scene takes place. Describe the physical location, setting, lighting, or any other environmental factors that affect the atmosphere or context of the video clip.
6. Temporal Context: Provide information about the timing of events within the scene. Describe the natural progression of time (e.g., morning, afternoon, evening) or any time-sensitive elements that contribute to the unfolding of the events.

Strict Requirements:

• Do not use generic descriptions, inferred names, or pronouns to refer to characters (e.g., "he," "they," "the man").
• The generated descriptions of the video clip should include every detail observable in the frames and mentioned in the audio dialogues. (**GPT-4o, Qwen2.5-VL-7b Variant:** • *The generated descriptions of the video clip should include every detail observable in the frames and mentioned in the audio dialogues.*)
• Pay close attention to any introduction of characters' names, titles, or other identifiers provided in the frames or audio.
• Whenever possible, include natural time expressions and physical location cues in the descriptions to improve contextual understanding. These should be based on inferred situational context (e.g., "in the evening at the dinner table," "early morning outside the building").
• Include relevant background, common knowledge and environmental factors when needed (e.g., location, weather, setting) to provide a fuller understanding of the context.

---

*(Continued on next page)*

• Maintain a natural, narrative flow in the description, ensuring that it reads like a coherent summary of the events in the video.
• Remember you are looking at key frames and audio dialogue information, not the full video, so focus on what can be observed from these specific materials. (***GPT-4o, Qwen2.5-VL-7b Variant:** • Remember you are looking at key frames and audio dialogue information, not the full video, so focus on what can be observed from these specific materials.*)

Example Output:

"As Margaret returns with the teapot, Tom stands up to help her pour the tea, gesturing politely as she hands him a cup. Margaret sits back down. Margaret leans forward slightly, her hands resting on the table, and after a moment of silence, she speaks again, her voice steady but filled with a hint of urgency. Tom listens closely, his brow furrowing slightly as he takes in her words. He responds quietly, nodding slowly as he processes the information."

**RAG Answer Prompt (GPT-4o)**

Based on the following video description, answer the question as concisely as possible. Provide only the direct answer without explanations or reasoning.

Question: {question}

Relevant Video Clip Captions: {retrived_clips}

Answer:

Table 24: The prompts for the experiments of the Socratic Models. For models that take either raw video (gemini-1.5-pro, Qwen2.5-Omni-7b) input or video frames with ASR transcripts (GPT4o, Qwen2.5-VL-7b), the description generation prompt has minor differences, which are indicated in italicized parentheses.

## M.3 PROMPTS FOR M3-AGENT

Table 25 shows the prompt used by Gemini-Agent and Gemini-GPT4o-Hybrid during memorization.

Table 26 shows the prompt used by Gemini-Agent and Gemini-GPT4o-Hybrid during control.

Table 27 shows the prompt used by M3-Agent during the control process. The system prompt at the beginning of each session specifies the overall task objectives. The instruction prompt appended at the start of each round provides the question and detailed guidance. The last-round prompt, used only in the final round, signals the agent that it is the final opportunity to respond.

**Memorization Prompt ( `memory-gemini-prompt`, `memory-7b-prompt` )**

You are given a video along with a set of character features. Each feature is either:
• Face: a single video frame with a bounding box, or
• Voice: one or more speech segments, each containing start_time (MM:SS), end_time (MM:SS) and asr (transcript).
Every feature has a unique ID enclosed in angle brackets (e.g. <face_1>, <voice_2>).

Your Tasks (produce both in the same response) :

1. Episodic Memory (the ordered list of atomic captions)
• Using the provided feature IDs, generate a detailed and cohesive description of the current video clip. The description should capture the complete set of observable and inferable events in the clip. Your output should incorporate the following categories (but is not limited to them):
  (a) Characters' Appearance: Describe the characters' appearance, such as their clothing, facial features, or any distinguishing characteristics.

*(Continued on next page)*

(b) Characters' Actions & Movements: Describe specific gesture, movement, or interaction performed by the characters.

(c) Characters' Spoken Dialogue: Quote—or, if necessary, summarize—what are spoken by the characters.

(d) Characters' Contextual Behavior: Describe the characters' roles in the scene or their interaction with other characters, focusing on their behavior, emotional state, or relationships.

2. Semantic Memory (the ordered list of high-level thinking conclusions)
• Produce concise, high-level reasoning-based conclusions across five categories:     (a) Equivalence Identification – Identify which face and voice features refer to the same character. Use the exact format: Equivalence: <face_x>, <voice_y>. Include as many confident matches as possible.

(b) Character-level Attributes – Infer abstract attributes for each character, such as: Name (if explicitly stated), Personality (e.g., confident, nervous), Role/profession (e.g., host, newcomer), Interests or background (when inferable), istinctive behaviors or traits (e.g., speaks formally, fidgets). Avoid restating visual facts—focus on identity construction.

(c) Interpersonal Relationships & Dynamics – Describe the relationships and interactions between characters: Roles (e.g., host-guest, leader-subordinate), Emotions or tone (e.g., respect, tension), Power dynamics (e.g., who leads), Evidence of cooperation, exclusion, conflict, etc.

(d) Video-level Plot Understanding – Summarize the scene-level narrative, such as: Main event or theme, Narrative arc or sequence (e.g., intro → discussion → reaction), Overall tone (e.g., formal, tense), Cause-effect or group dynamics.

(e) Contextual & General Knowledge – Include general knowledge that can be learned from the video, such as: Likely setting or genre (e.g., corporate meeting, game show), Cultural/procedural norms, Real-world knowledge (e.g., "Alice market is pet-friendly"), Common-sense or format conventions.

Strict Requirements (apply to both sections unless noted)

1. If a character has a provided feature ID, refer to that character only with the ID (e.g. <face_1>, <voice_2>).
2. If no ID exists, use a short descriptive phrase (e.g. "a man in a blue shirt").
3. Do not use "he," "she," "they," pronouns, or invented Names.
4. Keep face/voice IDs consistent throughout.
5. Describe only what is grounded in the video or obviously inferable.
6. Include natural Time & Location cues and setting hints when inferable.
7. Each Episodic Memory line must express one event/detail; split sentences if needed.
8. Output English only.
9. Output a Python list of sentences for each memory type.

Additional Rules for Episodic Memory

1. Do not mix unrelated aspects in one memory sentence.
2. Focus on appearance, actions/movements, spoken dialogue (quote or summary), contextual behavior.

Additional Rules for Semantic Memory

1. For Equivalence lines, use the exact format: Equivalence: <face_x>, <voice_y>.
2. Do not repeat simple surface observations already in the captions.
3. Provide only final conclusions, not reasoning steps.

Expected Output Format

Return the result as a single Python dict containing exactly two keys:

{

"episodic_memory": [

"In the bright conference room, <face_1> enters confidently, giving a professional appearance as he approaches <face_2> to shake hands.",

"<face_1> wears a black suit with a white shirt and tie. He has short black hair and wears glasses.",

"<face_2>, dressed in a striking red dress with long brown hair.",

"<face_2> smiles warmly and greets <face_1>. She then sits down at the table beside him, glancing at her phone briefly while occasionally looking up.",

"<voice_1> speaks to the group, 'Good afternoon, everyone. Let's begin the meeting.' His voice commands attention as the room quiets, and all eyes turn to him.",

"<face_2> listens attentively to <voice_1>'s words, nodding in agreement while still occasionally checking her phone. The atmosphere is professional, with the participants settling into their roles for the meeting.",

"<face_1> adjusts his tie and begins discussing the agenda, engaging the participants in a productive conversation."
],
"semantic_memory": [
"Equivalence: <face_1>, <voice_1>",
"<face_1>'s name is David.",
"<face_1> holds a position of authority, likely as the meeting's organizer or a senior executive.",
"<face_2> shows social awareness and diplomacy, possibly indicating experience in public or client-facing roles.",
"<face_1> demonstrates control and composure, suggesting a high level of professionalism and confidence under pressure.",
"The interaction between <face_1> and <face_2> suggests a working relationship built on mutual respect.",
"The overall tone of the meeting is structured and goal-oriented, indicating it is part of a larger organizational workflow."
]
}

Please only return the valid python dict (which starts with "{" and ends with "}") containing two string lists in "episodic_memory" and "semantic_memory", without any additional explanation or formatting.

Table 25: Memorization prompt for `memory-gemini-prompt` and `memory-7b-prompt`.

**Control Prompt**

You are given a question and some relevant knowledge about a specific video. You are also provided with a retrieval plan, which outlines the types of information that should be retrieved from a memory bank in order to answer the question. Your task is to reason about whether the provided knowledge is sufficient to answer the question. If it is sufficient, output [ANSWER] followed by the answer. If it is not sufficient, output [SEARCH] and generate a query that will be encoded into embeddings for a vector similarity search. The query will help retrieve additional information from a memory bank that contains detailed descriptions and high-level abstractions of the video, considering the question, the provided knowledge, and the retrieval plan.

Your response should contain two parts:
1. Reasoning
• Analyze the question, the knowledge, and the retrieval plan.
• If the current information is sufficient, explain why and what conclusions you can draw.
• If not, clearly identify what is missing and why it is important.
2. Answer or Search

• [ANSWER]: If the answer can be derived from the provided knowledge, output [ANSWER] followed by a short, clear, and direct answer.
• When referring to a character, always use their specific name if available.
• Do not use ID tags like <character_{1}> or <face_{1}>.
• [SEARCH]: If the answer cannot be derived yet, output [SEARCH] followed by a single search query that would help retrieve the missing information.

Instructions for [SEARCH] queries:
• Use the retrieval plan to inform what type of content should be searched for next. These contents should cover aspects that provide useful context or background to the question, such as character names, behaviors, relationships, personality traits, actions, and key events.
• Use keyword-based queries, not command sentences. Queries should be written as compact keyword phrases, not as full sentences or instructions. Avoid using directive language like "Retrieve", "Describe", or question forms such as "What", "When", "How".
• Keep each query short and focused on one point. Each query should target one specific type of information, without combining multiple ideas or aspects.
• Avoid over-complexity and unnecessary detail. Do not include too many qualifiers or conditions. Strip down to the most essential keywords needed to retrieve valuable content.
• The query should target information outside of the existing knowledge that might help answer the question.
• For time-sensitive or chronological information (e.g., events occurring in sequence, changes over time, or specific moments in a timeline), you can generate clip-based queries that reference specific clips or moments in time. These queries should include a reference to the clip number, indicating the index of the clip in the video (a number from 1 to N, where a smaller number indicates an earlier clip). Format these queries as "CLIP_x", where x should be an integer that indicates the clip index. Note only generate clip-based queries if the question is about a specific moment in time or a sequence of events.
• You can also generate queries that focus on specific characters or characters' attributes using the id shown in the knowledge.
• Make sure your generated query focus on some aspects that are not retrieved or asked yet. Do not repeatedly generate queries that have high semantic similarity with those generated before.

Example 1:

Input:
Question: How did the argument between Alice and Bob influence their relationship in the story?
Knowledge:
[
  {{
    "query": "What happened during the argument between Alice and Bob?",
    "related memories": {{
      "CLIP_2": [
        "<face_1> and <face_2> are seen arguing in the living room."
        "<face_1> raises her voice, and <face_2> looks upset."
        "<face_1> accuses <face_2> of not listening to her."
      ],
    }}
  }}
]

Output:
It seems that <face_1> and <face_2> are arguing about their relationship. I need to figure out the names of <face_1> and <face_2>.
[SEARCH] What are the names of <face_1> and <face_2>?

Example 2:

Input:
Question: How did the argument between Alice and Bob influence their relationship in the story?
Knowledge:
[
  {{
    "query": "What happened during the argument between Alice and Bob?",
    "related memories": {{
      "CLIP_2": [
        "<face_1> and <face_2> are seen arguing in the living room."
        "<face_1> raises her voice, and <face_2> looks upset."
        "<face_1> accuses <face_2> of not listening to her."
      ],
    }}
  }},
  {{
    "query": "What are the names of <face_1> and <face_2>?",
    "related memories": {{
      "CLIP_1": [
        "<face_1> says to <face_2>: 'I am done with you Bob!'",
        "<face_2> says to <face_1>: 'What about now, Alice?'"
      ],
    }}
  }}
]

Output:
It seems that content in CLIP_2 shows exactly the argument between Alice and Bob. To figure out how did the argument between Alice and Bob influence their relationship, I need to see what happened next in CLIP_3.
[SEARCH] What happened in CLIP_3?

Now, generate your response for the following input:

Question: {question}

Knowledge: {search_results}

Output:

**Control Prompt (last round)**

You are given a question about a specific video and a dictionary of some related information about the video. Each key in the dictionary is a clip ID (an integer), representing the index of a video clip. The corresponding value is a list of video descriptions from that clip.

Your task is to analyze the provided information, reason over it, and produce the most reasonable and well-supported answer to the question.

Output Requirements:
• Your response must begin with a brief reasoning process that explains how you arrive at the answer.
• Then, output [ANSWER] followed by your final answer.
• The format must be: Here is the reasoning... [ANSWER] Your final answer here.
• Your final answer must be definite and specific — even if the information is partial or ambiguous, you must infer and provide the most reasonable answer based on the given evidence.
• Do not refuse to answer or say that the answer is unknowable. Use reasoning to reach the best possible conclusion.

*(Continued on next page)*

Additional Guidelines:
• When referring to a character, always use their specific name if it appears in the video information.
• Do not use placeholder tags like <character_1> or <face_1>.
• Avoid summarizing or repeating the video information. Focus on reasoning and answering.
• The final answer should be short, clear, and directly address the question.

Input:
• Question: {question}
• Video Information: {search_results}

Output:

Table 26: Control prompt for Gemini-Agent and Gemini-GPT4o-Hybrid.

| system_prompt |
| --- |
| You are given a question and some relevant knowledge. Your task is to reason about whether the provided knowledge is sufficient to answer the question. If it is sufficient, output [Answer] followed by the answer. If it is not sufficient, output [Search] and generate a query that will be encoded into embeddings for a vector similarity search. The query will help retrieve additional information from a memory bank. |

Question:

| instruction_prompt |
| --- |
| Output the answer in the format: |
| Action: [Answer] or [Search] |
| Content: {content} |
| |
| If the answer cannot be derived yet, the {content} should be a single search query that would help retrieve the missing information. The search {content} needs to be different from the previous. |
| You can get the mapping relationship between character ID and name by using search query such as: "What is the name of <character_{i}>" or "What is the character id of {name}". |
| After obtaining the mapping, it is best to use character ID instead of name for searching. |
| If the answer can be derived from the provided knowledge, the {content} is the specific answer to the question. Only name can appear in the answer, not character ID like <character_{i}>. |

| last_round_prompt |
| --- |
| The Action of this round must be [Answer]. If there is insufficient information, you can make reasonable guesses. |

Table 27: The prompts used by M3-Agent during the control process.

