# OpenReview forum: "Seeing, Listening, Remembering, and Reasoning: A Multimodal Agent with Long-Term Memory"
_ICLR.cc/2026/Conference — ICLR 2026 Poster_

### Official Review · Reviewer_BEzR · 2025-10-26

**Soundness:** 2
**Presentation:** 3
**Contribution:** 3
**Rating:** 2
**Confidence:** 4

**Summary:**

This paper presents M3-Agent, a multimodal agent framework that integrates long-term memory with reinforcement fine-tuning. The system processes continuous visual and auditory inputs to form episodic and semantic memories, which are stored in an entity-centric multimodal graph.

To evaluate the effectiveness of memory effectiveness and memory-based reasoning in multimodal agents, the authors also build M3-Bench, a new benchmark for long-video question answering. It consists of two parts: M3-Bench-robot, containing 100 egocentric videos recorded from a robot’s viewpoint, and M3-Bench-web, including 920 web videos across diverse scenarios. Experiments show that M3-Agent surpasses baselines including GPT-4o and Gemini-1.5-pro–based prompting agents on M3-Bench and VideoMME-long.

**Strengths:**

1. This paper proposed a challenging and realistic video understanding benchmark M3-Bench that goes beyond existing works. Instead of relying solely on visual perception,  it further requires the agent to leverage higher-level cognitive abilities and integrate world knowledge to solve complex tasks that involve long-horizon dependencies, temporal reasoning, and cross-modal understanding. Moreover, M3-Bench fills an important gap in evaluating memory-based long-term multimodal reasoning, especially in scenarios with robot-perspective data.

2. This paper proposed a complete and well-engineered system M3-Agent that integrates video/audio perception, memory construction, and multi-step retrieval control. The overall framework is coherent and clear: episodic memory captures fine-grained observations, semantic memory provides additional cues for retrieval, and the entity-centric memory graph maintains information consistency across long contexts.

**Weaknesses:**

1. Lack of real methodological novelty. The overall pipeline resembles VideoAgent [1]: the agent first perceives multimodal inputs, then stores textualized or embedding-based observations into an external memory, performs RAG during query time, and finally conducts multi-step reasoning over the retrieved memory to produce answers. The entity graph is not novel, similar identity linking appears in StoryTeller [2]. The RL training also follows a conventional approach, similar to Search-R1 [3], making the proposed framework appear more like an incremental engineering integration rather than a fundamentally novel paradigm.

2. Memory quality is central to paper claims yet barely evaluated. The entity-based memory graph is central to the contribution, but its behavior over long horizons is not analyzed. There is no evaluation of memory corruption, contradiction resolution, or scalability as memory grows. Conflict handling is briefly mentioned via weighted voting (Sec. 4.1) but is under-specified and not measured experimentally.

3. The claim of being an "online agent" is unsubstantiated. Despite repeatedly claiming “online” and “streaming” capability, the paper does not report any latency, throughput, or memory scaling measurement, nor demonstrates real-time processing. The entire pipeline is evaluated offline on fixed-length videos, and no evidence is provided that M3-Agent is suitable for online deployment.

4. Insufficient Evaluation of “World Knowledge Construction”. The paper rightly points out that prior methods focus too much on low-level visual details and neglect high-level world knowledge (e.g., character identity, attributes). However, the evaluation on M3-Bench (with five QA types) does not sufficiently probe the system’s ability to generalize and transfer knowledge. For instance, if the agent observes in multiple videos that “green bins are for recycling,” can it apply this rule to a new household scene?

5. Failure Case Analysis Could Be More In-Depth. Appendix J includes a case study showing successes and “hard cases” but lacks quantitative error categorization. While the paper identifies types of failures (e.g., spatial reasoning), it doesn’t quantify how prevalent each error type is. For instance, what percentage of errors on M3-Bench-robot stem from spatial misunderstandings?  Without this, it’s hard to gauge the main bottlenecks.

[1] VideoAgent: A Memory-augmented Multimodal Agent for Video Understanding

[2] StoryTeller: Improving Long Video Description through Global Audio-Visual Character Identification

[3] Search-R1: Training LLMs to Reason and Leverage Search Engines with Reinforcement Learning

**Questions:**

1. What if one doesn’t have access to in-house data? Is your training data sourced from the same distribution as your benchmark? If so, could this create an unfair advantage for your method and negatively impact fair comparison in future work?
2. During RL training, you use GPT-4o for scoring. Does this reliance on API calls significantly increase training time or introduce practical bottlenecks?
3. What is the typical length of the input history? When retrieving memories, are both images and text included directly as inputs to the model? Is there any further compression or processing applied to the retrieved multimodal content?
4. The original Video-MME Long paper [1] reports a Gemini-1.5-Pro result of 73.6, which is notably higher than the numbers reported in your paper. Additionally, the baselines you compare against appear outdated. Have you evaluated your method against more recent state-of-the-art models?

[1] Video-MME: The First-Ever Comprehensive Evaluation Benchmark of Multi-modal LLMs in Video Analysis

---

> ### Author Response · Authors · 2025-11-19
>
> We sincerely thank you for your effort in reviewing our paper. We greatly appreciate your valuable comments and insightful feedback. We also appreciate your recognition of our contributions in  proposing the novel M3-Bench benchmark, detailed introduction for M3-Agent and M3-Bench, and comprehensive experiments. In our response, we will address each question individually, quoting them and providing our answer accordingly.
>
> > **Q1: Lack of real methodological novelty. The overall pipeline resembles VideoAgent [1]: the agent first perceives multimodal inputs, then stores textualized or embedding-based observations into an external memory, performs RAG during query time, and finally conducts multi-step reasoning over the retrieved memory to produce answers. The entity graph is not novel, similar identity linking appears in StoryTeller [2]. The RL training also follows a conventional approach, similar to Search-R1 [3], making the proposed framework appear more like an incremental engineering integration rather than a fundamentally novel paradigm.**
>
> A1: The core methodology contribution of M3-Agent is its solution to the consistency problem in multimodal long-term memory, which is a fundamental bottleneck for building reliable multimodal agents that must continuously process streaming inputs. Existing works such as VideoAgent, StoryTeller and Search-R1 do not address this challenge. Long-term memory consistency requires an agent to maintain coherent, non-contradictionary representations of the same entities as they appear and reappear over extended periods. To overcome this challenge, M3-Agent introduces several key methodological contributions:
> 1. Construct entity-centric long-term memory for multimodal agents, which enables the agent to accumulate world knowledge and build a more effective (capturing more useful information) and consistent (maintaining identity and attribute coherent over time) memory.
> 2. Multimodal streaming memorization. M3-Agent processes continuously perceives streaming multimodal inputs and generates both episodic and semantic memory.
> 3. Design parallel memorization and control processes and the full system can be optimized by RL. Experimental results show that, after training, M3-Agent outperforms all baselines.
>
> Besides above contributions, M3-Agent also differs from prior works in many key aspects. VideoAgent primarily depends on prompt engineering, while M3-Agent is trained by RL, enabling an agent based on small-size open-source model to outperform prompt-based agent built on prompting proprietary LLMs (see Table 4). StoryTeller focuses mainly on person identity tracking, whereas M3-Agent builds a comprehensive entity graph, not only people identity, but also their features, priors and relationships. In addition, StoryTeller's Global Decoding approach requires full video access, making it unsuitable for streaming settings. In contrast, M3-Agent directly learns to unify voice_id and face_id and applies conflict-resolution mechanism, enabling robust, real-time identity consistency. Finally, although Search-R1 and M3-Agent both use RL to optimize multi-turn reasoning, the problem settings, objectives, and architectures are fundamentally different. The presence of RL in both should not diminish the novelty of the M3-Agent framework.

---

> ### Author Response · Authors · 2025-11-19
>
> > **Q2: Memory quality is central to paper claims yet barely evaluated. The entity-based memory graph is central to the contribution, but its behavior over long horizons is not analyzed. There is no evaluation of memory corruption, contradiction resolution, or scalability as memory grows. Conflict handling is briefly mentioned via weighted voting (Sec. 4.1) but is under-specified and not measured experimentally.**
>
> A2: Thanks for your feedback. We appreciate your recognition that the entity-based memory graph is a contribution. According to your advice, we provide a more detailed analysis of memory.
>
> - Memory Quality. In Appendix G Evaluation of Memorization, we report our evaluation of memory quality. Specifically, we adapt AutoDQ-P, AutoDQ-R and AutoDQ-F1 to assess the descriptive memory, and we use Precision, Recall and F1 to assess cross-modal identity mapping. The results shown that after imitation learning, memory-7b-sft improves AutoDQ-F1 by +21.3% and identity mapping F1 by +70.1% compared with the untrained Qwen2.5-omni-7b. Compared to Gemini-1.5-pro, memory-7b-sft achieves +2.1% higher AutoDQ-F1 and +25.1% higher identity mapping F1.
>
> - Contradiction Resolution. We perform an ablation study by removing the voting mechanism from the long-term memory. Without the voting mechanism, the agent's QA accuracy drops by 3.2% on M3-Bench-robot, 2.5% on M3-Bench-web, and 4.5% on Video-MME-Long, demonstrating the necessary of the voting mechanism for contradiction resolution. The results have been updated in the revised paper (see **Table 17**).
>
> - Scalability as Memory Grows. To evaluate scalability, based on the question timestamps in M3-Bench-robot, we separately plot (1) the number of memory items versus video duration and (2) the cumulative QA accuracy versus video duration, as shown in Figure 5 (a) in the revised paper. The results show that as memory grows, QA accuracy improves slightly, indicating that the system maintains robust performance with increasing memory. Furthermore, after RL training, M3-Agent consistently outperforms prompting-based agents using Gemini-1.5-Pro and GPT-4o across all memory sizes and video durations.
>
> > **Q3: The claim of being an "online agent" is unsubstantiated. Despite repeatedly claiming “online” and “streaming” capability, the paper does not report any latency, throughput, or memory scaling measurement, nor demonstrates real-time processing. The entire pipeline is evaluated offline on fixed-length videos, and no evidence is provided that M3-Agent is suitable for online deployment.**
>
> A3: Thank you for raising this point. M3-Agent supports streaming video processing through a sliding window manner that operates on k-second clips (k=30s in our experiments). Specifically, for a 30-second sliding window, the performance is as follows (on a single GPU with 80GB):
>
> |Sliding-window size|Tool Call Latency|Model Inference Latency|Generated Memory Tokens|
> |:-:|:-:|:-:|:-:|
> |30s|9.6s|1.9s|696.4|
>
> Note that we have not yet focused on extensive system-level optimization (e.g., inference acceleration and parallelization). Therefore, there remains potential for further reducing latency in real-world deployments through engineering optimization. Here, we use the term "online" to emphasize that M3-Agent can process 30-second videos via streaming and generate consistent memories, without the need to accumulate longer video segments. Moreover, memories can be generated and used for subsequent retrieval within one minute.

---

> ### Author Response · Authors · 2025-11-19
>
> > **Q4: Insufficient Evaluation of “World Knowledge Construction”. The paper rightly points out that prior methods focus too much on low-level visual details and neglect high-level world knowledge (e.g., character identity, attributes). However, the evaluation on M3-Bench (with five QA types) does not sufficiently probe the system’s ability to generalize and transfer knowledge. For instance, if the agent observes in multiple videos that “green bins are for recycling,” can it apply this rule to a new household scene?**
>
> A4: Thanks for your question. We agree that assessing knowledge generalization and transfer is essential. In fact, many M3-Bench questions naturally fall into this category. For example:
>
> - [video_id: bedroom_eei43] Question: Where should the yellow coat go?
> At the beginning of the video, Ella mentions that thick clothes should be placed on the upper layer of the wardrobe. To answer the question correctly, the agent must (1) recognize that the yellow coat is thick and (2) recall and apply the general rule that thick clothes belong on the upper layer.
> - [video_id: kitchen_eei21] Question: How to deal with the cup Alice using to drink orange juice?
> Early in the video, Micky tells Alice that used cups should be placed on the top shelf of the cabinet. Answering correctly requires extracting this general rule and applying it to the current situation.
>
> To quantify this, we provide GPT-4o with the question and human-labeled reasoning paths and prompt it to identify which questions involve knowledge generalization and transfer. In total, 16.29% of M3-Bench-robot questions and 13.10% of M3-Bench-web questions fall into this category.
>
> The label prompt:
>
> ```
> Based on the given question and reason, determine whether the question is aimed at assessing the ability to generalize and transfer knowledge. Only output YES or NO.
>
> Question: {}
> Reason: {}
> Answer:
> ```
>
> > **Q5: Failure Case Analysis Could Be More In-Depth. Appendix J includes a case study showing successes and “hard cases” but lacks quantitative error categorization. While the paper identifies types of failures (e.g., spatial reasoning), it doesn’t quantify how prevalent each error type is. For instance, what percentage of errors on M3-Bench-robot stem from spatial misunderstandings? Without this, it’s hard to gauge the main bottlenecks.**
>
> A5: We performed a case study of M3-Bench, covering both M3-Bench-web and M3-Bench-robot. Specifically, we sampled 50 failure cases from M3-Agent's responses and examined the underlying causes. The errors can be categorized as follows.
>
> |Error Type|Percentage|Explainataion|
> |:-|:-:|:-|
> |Reasoning about fine-grained details|50%|The agent fails to extract precise, detailed information from its observations. (See examples in Appendix L)|
> |Spatial reasoning|20%|The agent fails to understand spatial layout or tracking spatial changes. (See examples in Appendix L)|
> |Incorrect search strategy |12%|The agent does not correctly decompose the query or fails to execute an effective search plan to gather necessary information.|
> |Missing Human names|10%|The agent fails to extract the relevant character names required to answer the question.|
> |Others|8%||
>
> > **Q6: What if one doesn’t have access to in-house data? Is your training data sourced from the same distribution as your benchmark? If so, could this create an unfair advantage for your method and negatively impact fair comparison in future work?**
>
> A6: Our training data comes from a different distribution than M3-Bench, and there is no overlap in video sources. This ensures that our method does not benefit from any benchmark leakage or unfair advantage. To promote transparency, reproducibility and to allow the community to verify this, we will release our models, code, and the training datasets used during RL training.
>
> In addition, we would like to emphasize that our approach is designed to be general, rather than tailored to any particular benchmark. To demonstrate this point, we also evaluate our approach on VideoMME-Long and observe consistent strong performance. By releasing the code and models, we expect to provide a general agent and further demonstrate its broad effectiveness.
>
> > **Q7 : During RL training, you use GPT-4o for scoring. Does this reliance on API calls significantly increase training time or introduce practical bottlenecks?**
>
> A7: Using GPT-4o for scoring does not increase training time or introduce bottleneck. Under our training setup (see Table 14), each experiment requires approximately 128 queries per minute to the GPT-4o API. The average response latency for receiving scores is 0.62 seconds, which is negligible compared to the overall RL optimization cycle (rollout latency is 206.90 seconds, gradient updates latency is 8.69 seconds).

---

> ### Author Response · Authors · 2025-11-19
>
> > **Q9: The original Video-MME Long paper [1] reports a Gemini-1.5-Pro result of 73.6, which is notably higher than the numbers reported in your paper. Additionally, the baselines you compare against appear outdated. Have you evaluated your method against more recent state-of-the-art models?**
>
> A9: Thanks for the question. Our evaluation setting is fundamentally different with the original Video-MME paper. The original Video-MME paper allows model to access the entire video and question simultaneously, a typical evaulation setting for offline VQA. In contrast, our work targets streaming video, where the model processes long videos clip by clip, without re-accessing the full history. This setting better reflects real-world constraints, where revisiting hours or days long video for every query is computationally impractical.
>
> Regarding baselines, we have made effort to include strong and up-to-date methods capable of handling streaming inputs. The strongest baseline is prompting agent (M3-Agent based on prompting Gemini-1.5-Pro and GPT-4o). For Socratic Model baselines, we include both proprietary (GPT-4o, Gemini-1.5-Pro) and open-source multimodal models (Qwen2.5-VL, Qwen2.5-Omni). For online video understanding methods, we compare with the latest available open-source models, including MovieChat, MA-LLM, and Flash-VStream.
>
> **Thank you again for your valuable comments and feedback. In addition to the responses above, we have revised the paper accordingly. Please let us know if our responses address your concerns. We sincerelly appreciate your time and effort in helping us enhance our work.**

---

> > ### Comment · Reviewer_BEzR · 2025-11-25
> > **I appreciate the contributions of this paper, and I will raise my score.**
> >
> > Thank you for the detailed responses and the additional experiments.
> >
> > For **Q1**, I think your explanation makes sense, and it would be great if these comparisons could be added to later versions of the paper.
> >
> > For **Q2**, the reply is quite thorough; the only part that still feels a bit missing is the discussion on memory corruption.
> >
> > For **Q3** and **Q4**, I feel the answers are sufficient.
> >
> > For **Q5**, it seems that the major errors(the main bottleneck) come from visual understanding. Since M3-Bench is mainly meant to evaluate consistency, I think this point could use a bit more clarification.
> >
> > For **Q6** and **Q7**, I don’t have any issues.
> >
> > For **Q9**, I still have some questions. If VideoAgent always needs to access the full history, does that make the computational cost much higher? And does your method aim to strike a balance between cost and performance? Is there any comparison under the same constraints?
> >
> > Overall, I appreciate the contributions of this paper, and I will raise my score.

---

> ### Author Response · Authors · 2025-11-26
>
> We really appreciate your thoughtful feedback and recognition of our work's contributions. We are grateful for the improved score, which is truly encouraging. Your constructive feedback helped us enhance our work, and we have revised our paper accordingly.
>
> ---
> We apologize for inadvertently omitting our response to Q8 in the previous version. We provide the answer below.
> > **Q8: What is the typical length of the input history? When retrieving memories, are both images and text included directly as inputs to the model? Is there any further compression or processing applied to the retrieved multimodal content?**
>
> A8: When retrieving memories, both episodic and semantic text memories are appended directly to the model's context without additional compression or processing. On the M3-Bench, a full trajectory (may involve multiple search turns) contains an average of 3317 tokens, of which around 2001 tokens come from the retrieved memory content.
>
> ---
> **Response to other feedback:**
>
> > **For Q1, I think your explanation makes sense, and it would be great if these comparisons could be added to later versions of the paper.**
>
> We have revised the paper to include these comparison in Section 2 Related Work and add a citation to SearchR1 in Line 71 of the Introduction.
>
> > **For Q2, the reply is quite thorough; the only part that still feels a bit missing is the discussion on memory corruption.**
>
> Regarding memory corruption, memories associated with frequently activated entities are repeatedly revisited during task execution, which provides the agent opportunities to correct inaccurate or outdated information through the contradiction resolution mechanism. Memories related to rarely activated entities have relatively limited influence on task execution. Therefore, the contradiction resolution mechanism helps reduce the effects of memory corruption.
>
> > **For Q5, it seems that the major errors(the main bottleneck) come from visual understanding. Since M3-Bench is mainly meant to evaluate consistency, I think this point could use a bit more clarification.**
>
> Thanks for your constructive suggestion. We have revised our paper to include a more detailed analysis of hard cases in M3-Bench (see Appendix L Case Study).
>
> Our analysis shows that the main bottlenecks are reasoning about fine-grained details (50%) and spatial reasoning (20%). Some errors are indeed related to visual understanding. In addition, we should note that a critical issue is that the agent often fails to realize which information should be extracted and memorized. In particular, when the model is provided with both the video and the question, it can generate the correct answer. However, when only the video is provided, the agent must first decide what to memorize based on its estimation of importance. This process may overlook information that is crucial for the future questions. Simply encouraging the agent to generate more details is impractical as it can lead to cognitive overload. Instead, the agent should adopt more effective attention and selective memorization mechanisms, allowing it to retain task-relevant details. We leave this direction for future work.

---

> ### Author Response · Authors · 2025-11-26
>
> > **For Q9, I still have some questions. If VideoAgent always needs to access the full history, does that make the computational cost much higher? And does your method aim to strike a balance between cost and performance? Is there any comparison under the same constraints?**
>
> Thanks for your question. For long videos, there are typically two common frame sampling strategies: (1) Use a fixed number of frames, regardless of video length. For example, Qwen-2.5-VL/-Omni uniformly samples up to 768 frames [1]. (2) Use a fixed sampling fps. For example, Gemini-1.5-Pro defaults to 1 fps [2].
>
> In a streaming video agent scenario, such as a robot continuously observing its environment for hours or even days, if the entire video history must be accessed to answer a question, neither approaches is practical. With a fixed maximum number of frames, the sampling rate decreases as the video grows longer, increasing the probability that important frames are missed. With a fixed fps, the number of input tokens grows linearly with time and eventually exceeds the maximum input length. As a result, in streaming scenarios, it is not feasible to always access to the full history.
>
> In M3-Agent, the memorization process continuously processes a streaming video clip by clip and maintains a multimodal long-term memory. When a question is raised, the control process is invoked, and the search function calls require a constant computational cost. Our goal is to support  the streaming video agent scenario while improving overall performance under this constraint.
>
> To quantitatively demonstrate above discussion, we split the Video-MME-Long into the shortest 30% and longest 30% of videos based on their duration, and measured both the total token usage and accuracy for Qwen2.5-Omni, Gemini-1.5-Pro, and M3-Agent. (For Qwen2.5-Omni and Gemini-1.5-Pro, we follow the original Video-MME-Long paper where both the full video and the question are provided as input.)
>
> |||shortest 30%|longest 30%|
> |:-:|:-:|:-:|:-:|
> |average video length||1834.21s|3389.96s|
> |Qwen2.5-Omni|token usage|114513.6|114211.7|
> ||QA accuracy|0.589|0.511|
> |Gemini-1.5-Pro|token usage|519157.2|957970.9|
> ||QA accuracy|0.778|0.622|
> |M3-Agent-control|token usage|2726.8|2845.6|
> ||QA accuracy|0.600|0.611|
>
> The results show that Qwen2.5-Omni maintains similar token usage, but its accuracy drops by 7.8% for longer video. Gemini-1.5-Pro exhibits accuracy drop of 15.6% (likely because model struggles with long context [3,4]), and its token usage increases significantly for longer videos. In contrast, **M3-Agent maintains stable token usage and stable accuracy**, even as video length increases.
>
> [1] https://github.com/QwenLM/Qwen3-VL/blob/main/qwen-vl-utils/src/qwen_vl_utils/vision_process.py#L144
> [2] https://ai.google.dev/gemini-api/docs/video-understanding
> [3] Lost in the Middle: How Language Models Use Long Contexts, TACL 2024.
> [4] NoLiMa: Long-Context Evaluation Beyond Literal Matching, ICML 2025.
>
> ---
> Thanks again for taking the time to engage in this discussion. We appreciate your time and effort in helping us improve our work. If you have any further questions or concerns, we would be happy to address them in the discussion period.

---

### Official Review · Reviewer_urmt · 2025-10-31

**Soundness:** 2
**Presentation:** 3
**Contribution:** 3
**Rating:** 4
**Confidence:** 4

**Summary:**

This paper proposes M3-Agent, a multimodal agent architecture with long-term, entity-centric memory. The system runs two concurrent loops: (i) memorization that parses streaming video/audio into episodic and semantic memories; and (ii) control, which performs multi-turn retrieval and reasoning over this memory to answer questions. To evaluate memory-based reasoning beyond low-level perception, this paper introduces M3-Bench, a long-video QA benchmark with 100 robot-egocentric videos and 920 web videos, annotated with question types targeting multi-evidence, multi-hop, cross-modal, and general knowledge extraction. Across M3-Bench and VideoMME-long benchmark, M3-Agent outperforms baselines.

**Strengths:**

1. M3-Bench is a novel evaluation benchmark designed to assess memory-based reasoning beyond low-level perception. This dataset comprises 100 robot-egocentric videos and 920 web videos, offering a good option for agent evaluation. The data processing and evaluation reflect the author's significant efforts.

2. This paper provides a detailed account of the development process and implementation specifics for constructing M3-bench and M3-agent, supported by comprehensive experiments. Furthermore, the paper features a well-structured layout, good presentation, and is easy to follow.

**Weaknesses:**

1. Automatic grading uses GPT-4o to judge correctness; although this paper report 96% agreement with human majority on a small set, relying on a single proprietary LLM as an oracle can introduce bias and inflate or depress certain methods’ scores.

2. GPT-4o was employed for data generation, evaluation, and baseline comparison. On one hand, this diminishes the paper's contribution, making it more akin to prompt engineering using GPT-4o. On the other hand, constructing data and conducting evaluations with the same model introduces preference bias.

3. The proposed M3 agent lacks technological innovation. From data processing to model training (DAPO), it relies entirely on existing methods, while its memory construction primarily depends on MLLMs.

**Questions:**

1. Beyond the 100-sample spot-check, did you run a larger human evaluation or cross-evaluator check (e.g., Claude/LLama-judge) to ensure conclusions are not tied to GPT-4o’s preferences?

2. How does index size, retrieval time, and accuracy scale as video hours and the number of identities grow?

3. The ablations show big drops without semantic memory and identity equivalence. Could you further decompose semantic memory into subtypes (attributes vs. relations vs. rules) to identify which contributes most by question type?

---

> ### Author Response · Authors · 2025-11-19
>
> We sincerely thank you for your effort in reviewing our paper. We greatly appreciate your valuable comments and insightful feedback. We also appreciate your recognition of our contributions in  proposing the novel M3-Bench benchmark, detailed introduction for M3-Agent and M3-Bench, and comprehensive experiments. In our response, we will address each question individually, quoting them and providing our answer accordingly.
>
> > **Q1: Automatic grading uses GPT-4o to judge correctness; although this paper report 96% agreement with human majority on a small set, relying on a single proprietary LLM as an oracle can introduce bias and inflate or depress certain methods’ scores.**
>
> A1: Thank you for raising this point. To further strengthen the robustness of our evaluation, we additionally use Gemini-1.5-Pro and Qwen3-32B to re-evaluate all experimental results in Table 4 (using the same prompt as for GPT-4o, as shown in Table 22). This cross-model evaluation leads to the same conclusion as the GPT-4o automated evaluator. The main results evaluated by Gemini-1.5-Pro and Qwen3-32B are shown in **Table 15** and **Table 16** respectively.
>
> In addition, when designing M3-Bench, we specifically considered the need to support feasible and reliable automated evaluation. During the construction of M3-Bench, annotators were explicitly required to produce question-answer pairs that support objective, verifiable evaluation. As specified in Appendix C.1, annotators must avoid overly open-ended, compound questions, or questions with multiple equally valid answers. The annotation guidelines make the questions challenging to answer but easy to evaluate, thereby enabling reliable automated evaluation. That is why the automated evaluator can achieve a high agreement rate of 96% with human majority judgments.
>
> > **Q2: GPT-4o was employed for data generation, evaluation, and baseline comparison. On one hand, this diminishes the paper's contribution, making it more akin to prompt engineering using GPT-4o. On the other hand, constructing data and conducting evaluations with the same model introduces preference bias.**
>
> A2: Thanks for bringing this to our attention. We would like to provide the following clarification.
>
> - GPT-4o for data generation: For data generation, we use GPT-4o and Gemini-1.5-Pro, in a mutually supervising and complementary manner, to synthesize partial imitation-learning data for the memorization policy. Our main contributions in training are not prompt engineering, but: (1) We perform imitation learning for memorization policy and reinforcement learning for control policy. The training of control policy does not use any synthetic data. (2) We propose a new algorithm that automatically annotates face-voice mappings for arbitrary long videos (see Appendix F.2). These data are used to train the memorization policy via imitation learning.
> - GPT-4o as automated evaluator: We use GPT-4o as an automated evaluator. To address potential bias, in our response to Q1, we additionally report evaluation results using Gemini-1.5-Pro, Qwen3-32b, which confirms the robustness of our findings.
> - GPT4o as baselines: We compare against three categories of baselines: (1) Socratic Method: including open-source models (Qwen2.5-VL, Qwen2.5-Omni) and proprietary models (Gemini-1.5-Pro, GPT-4o). (2) Online video understanding methods: MovieChat, MA-LMM and Flash-VStream, all implemented with open-sourced foundation models. (3) M3-Agent based on prompt engineering. We consider (3) especially important baselines, and in practice the strongest baselines, because they represent the most competitive alternative to our method that does not require additional training.
>
> In summary, GPT-4o is used only as a tool in synthesizing partial training data for memorization and as an automated evaluator. We choose GPT-4o as baseline because it is among the strongest closed-source models. Our main contribution, M3-Agent, is an RL-trained agent built on small foundation models. The results show that after RL training, M3-Agent based on small models can outperform prompting agents based on large proprietary models, highlighting the value of our training.

---

> ### Author Response · Authors · 2025-11-19
>
> > **Q3: The proposed M3 agent lacks technological innovation. From data processing to model training (DAPO), it relies entirely on existing methods, while its memory construction primarily depends on MLLMs.**
>
> A3: The key technological innovation of the M3-Agent framework includes:
>
> 1. We propose a novel framework for multimodal agent with long-term memory, enabling agents to continuously receive streaming multimodal inputs, gradually build world knowledge and form structured and consistent long-term memory. Maintaining consistency in multimodal long-term memory is significantly more challenging than in text-only memory, and existing MLLM-based approaches have not effectively addressed this problem. To overcome this bottleneck, we introduce a general framework and demonstrate its effectiveness through extensive experiments on multiple benchmarks.
> 2. M3-Agent constructs entity-centric long-term memory that allows the agent to accumulate world knowledge. This mechanism not only resolves key consistency issues, maintaining identity and attribute coherent over time, but also greatly expands the utility of memory, enabling the agent to answer high-level questions (e.g. about understanding of people and environment) more accurately.
> 3. M3-Agent formulates memorization and control as parallel processes and the full system can be optimized by RL. Experimental results show that, after training, M3-Agent based on small models can outperform prompting agents based on large proprietary models.
>
> > **Q4: Beyond the 100-sample spot-check, did you run a larger human evaluation or cross-evaluator check (e.g., Claude/LLama-judge) to ensure conclusions are not tied to GPT-4o’s preferences?**
>
> A4: As shown in A1, we have used Gemini-1.5-Pro and Qwen3-32b as independent cross-evaluators to verify the main results in Table 4. (We select Gemini-1.5-Pro because Claude is not accessible in our region, and we select Qwen3-32b as open-source evaluator because its stronger performance compared to LLaMA on most public benchmarks.)
>
> > **Q5: How does index size, retrieval time, and accuracy scale as video hours and the number of identities grow?**
>
> A5: Thanks for your question. These analyses are indeed helpful for understanding agent's behavior.
> - Impact of video duration. Based on the question timestamps in M3-Bench-robot, we separately plot memory size and QA accuracy versus video duration, as shown in **Figure 5 (a)** in the revised paper. We can observe that memory size grows approximately linearly with video duration. QA accuracy increases slightly as videos become longer, indicating that the system maintains robust performance even as memory grows. Furthermore, M3-Agent consistently outperforms the strongest baselines, prompting-based agents using Gemini-1.5-Pro and GPT-4o across all video durations.
> - Impact of identity count. Since M3-Bench-web provides a wide range of identity counts while maintaining similar video lengths (controlling for video length is necessary since it is the dominant factor influencing memory size), we plot memory size and accuracy against the number of identities on this dataset. The results are shown in **Figure 5 (b)** in the revised paper. We observe that memory size and QA accuracy are nearly unaffected by the number of identities. Notably, M3-Agent consistently surpasses the strongest baselines across all identity-count categories. For videos with many identities, the performance gap widens more, demonstrating that M3-Agent is more robust than baseline methods when handling more complex scenarios.
> - Retrieval time. Given the total number of memory items in our experiments, retrieval latency is dominated by the text-to-embedding service rather than the embedding-similarity computation. Thus, variations in video duration or identity count have negligible influence on retrieval time. In practice, the average latency is 0.5 seconds and the maximum is 1.6 seconds.
>
> > **Q6: The ablations show big drops without semantic memory and identity equivalence. Could you further decompose semantic memory into subtypes (attributes vs. relations vs. rules) to identify which contributes most by question type?**
>
> A6: Thank you for your question. To analyze how different types of semantic memory affect agent performance, we first utilize GPT-4o to label each memory item as an attribute, relation, general rule, or other. We then conduct an ablation study by removing each category in turn and measuring the resulting change in QA accuracy. The results listed in **Table 17** indicate that attribute, relation and rule memories all contribute to ahcieving the best performance.
>
> **Thank you again for your valuable comments and feedback. In addition to the responses above, we have revised the paper accordingly. Please let us know if our responses address your concerns. We sincerely appreciate your time and effort in helping us enhance our work.**

---

> > ### Comment · Reviewer_urmt · 2025-11-24
> >
> > I appreciate the authors’ response. Most of my previous concerns have been addressed. However, my main remaining concern is that the core contribution of this work lies in M3-Bench, whereas the M3 agent itself appears to lack sufficient technical novelty. This issue was also pointed out by Reviewer BEzR. In the rebuttal, the authors argue that the construction of long-term memory is a key contribution of the M3 agent. However, long-term memory itself is not a novel concept; the paper should more clearly and thoroughly explain how the proposed long-term memory mechanism differs from existing approaches and how it impacts the performance of the M3 agent. As a kind suggestion, I recommend visualizing the architecture of the M3 agent to help readers better understand its technical innovations.

---

> ### Author Response · Authors · 2025-11-25
>
> Many thanks for your valuable feedback and constructive suggestions. Following your suggestion, we have updated the M3-Agent architecture figure (**Figure 1**) to better emphasize its contribution. We clarify our contributions below.
>
> Unlike text-based agent, long-term memory in multimodal agent faces unique challenges: maintaining consistency over time and across modalities. These challenges have not been adequately addressed by existing methods. Our work tackles this key issue by **proposing an entity-centric memory framework that can accumulate world knowledge to form a coherent memory**.
>
> To further demonstrate its significance, we provide a qualitative comparison of entity-centric memory built by M3-Agent with standard memory generated by Gemini-1.5-Pro and GPT-4o, as shown in **Table 18** in the revised paper. We can observe that: (1) M3-Agent maintains consistent entity identities. Even when speakers' faces never appear on screen, M3-Agent correctly distinguish them and associate their voices with \<character_0\> or \<character_16\> in long-term memory. In contrast, Gemini-1.5-Pro describes them only as "an off-screen voice" and cannot distinguish between them, while GPT-4o fails to assign any consistent identity; (2) M3-Agent generates semantic memory in addition to episodic memory, effectively expanding the utility of the memory. For example, answering "Is Lucas skilled as cooking?" can not rely on episodic memory alone, as relevant events cannot be retrieved by key words such as *Lucas* or *cooking skills*. However, the semantic memory "\<character_23\> is named Lucas; \<character_23\> demonstrates a willingness to learn and improve cooking skills" is retrievable and directly supports correct answer.
>
> Quantitatively, out ablation studies show that removing semantic memory leads to a performance drops of **17.1%**, **19.2%** and **13.1%** on M3-Bench-robot, M3-Bench-web and Video-MME-Long, respectively. Following your suggestions, we further ablated different types of semantic memory, showing that all semantic memory types are essential.
>
> Besides long-term memory, we also emphasize our contribution to **system design**. We propose a new multimodal agent framework, consisting of parallel memorization and control processes, with explicitly defined interactions with long-term memory. The full system can be optimized by reinforcement learning. Our experiments show that M3-Agent significantly outperforms all baselines on M3-Bench-robot, M3-Bench-web and Video-MME-Long. Ablation studies further highlight the indispensability of entity-centric memory, multi-turn reasoning and search, and RL training in achieving the agent’s strong performance.
>
> At last, for clarity, we compare M3-Agent with existing related works in the accompanying table. M3-Agent stands out as the pioneering work that incorporates entity-centric multimodal memory and trains the entire agent system by RL.
>
> |Method|Multimodal Memory|Entity-Centric Memory|Contradiction Resolution|Infinite Video Support|Trainable/Optimizable|Parallel Process|
> |:-:|:-:|:-:|:-:|:-:|:-:|:-:|
> |Socratic Methods|X|X|X|√|X|X|
> |MovieChat|-|-|-|X|√|X|
> |MA-LMM|-|-|-|X|√|X|
> |Flash-VStream|-|-|-|X|√|X|
> |VideoAgent|√|X|X|√|X|X|
> |StoryTeller|-|-|-|X|√|-|
> |SearchR1|-|-|-|-|√|-|
> |M3-Agent|√|√|√|√|√|√|
>
> ------
> Please let us know if our replies address your concerns. Thanks for taking the time to engage in this discussion. We appreciate your time and effort in helping us improve our work.

---

### Official Review · Reviewer_2Awb · 2025-11-01

**Soundness:** 3
**Presentation:** 4
**Contribution:** 4
**Rating:** 8
**Confidence:** 4

**Summary:**

This paper introduces an agent (M3-agent) to process streams of video and audio input. In addition, the paper introduces a dataset and bechmark to evaluate long-horizon video understanding (M3-Bench).

The agent contains two sub-agents; one to build a semantic and episodic memory from long video streams, and another to query the streams to answer questions. The memory forming policy is trained behavior cloning/SFT, and the control policy is trained using RL (GRPO or DAPO).

The results across M3-Bench and Video-MME-Long are compelling; outperforming (1) modular systems that use proprietary models from Google and OpenAI with RAG, (2) off-the-shelf video understanding models, (3) Socratic models

The benchmark contains two components: one scraped from youtube and another of ecocentric videos collected by humans. The benchmark is designed so that the questions require information from video; not only audio. Many of the details about collection are provided in the paper and supplementary.

**Strengths:**

The paper provides both a practical recipe for a long-term memory, as well as a dataset and benchmark. In my opinion, it is an excellent contribution.

The paper is well-written and easy to follow; despite there being quite a lot of work being described.

The memory agent is interesting and flexible;  involves tool use -- e.g. to do facial recognition and associate memories with the speaker.

The proposed agent serves as an excellent baseline on the proposed benchmark. The fact that DAPO on the control policy improves over modular baselines helps demonstrate a potential path for improvement.

**Weaknesses:**

Overall I don't see major weaknesses; some minor ones about the memory (which I see as a baseline on the new benchmark).

* The control policy is trained using RL, but memory uses only SFT. Though it may be tricky to implement from an engineering perspective, RL seems especially useful also for the memory policy, too.
* Due to the method of constructing of entities in the memory dictionary, it seems speaker faces must be visible

**Questions:**

* Is the memory agent able to merge or edit memories?
* Is the agent able to call tools on its own, or is all tool use scripted beforehand?

**Details Of Ethics Concerns:**

I don't anticipate issues; just noting the submission involves these elements and that I don't have the background to review the practices used for data collection:
* scraping youtube
* maintaining privacy of the people in the youtube videos
* human subject doing video collection; these or other people may appear in videos

---

> ### Author Response · Authors · 2025-11-19
>
> We genuinely appreciate your recognition of our contributions and your positive feedback, which is truly encouraging. Thank you for the time you invested in reviewing our paper and for your insightful questions. In our response, we will address each question individually, quoting them and providing our answers accordingly.
>
> > **Q1: The control policy is trained using RL, but memory uses only SFT. Though it may be tricky to implement from an engineering perspective, RL seems especially useful also for the memory policy, too.**
>
> A1: Thanks for the insightful comment. We agree that RL has the potential to improve the memory policy. However, applying RL in memory generation comes with two key constraints:
>
> - Reward design is fundamentally hard. Defining a reward that balances correctness, detail, continuity (especially for episodic memory), and usefulness is difficult. And policies may still hack it, for example, by producing trivial but consistently correct outputs or by avoiding important details. Creating an effective reward that avoids these issues remains a major challenge.
> - Missing information feedback is hard to express. In rollout trajectories, it is difficult to indicate which important pieces of information are missing. Even if we design rewards that measure recall of key content, they provide only a single scalar score rather than identifying specific contents, making credit assignments extremely challenging. Training a model to accurately recover missing content may require substantial rollout computation and sophisticated rollout methods.
>
> Despite these challenges, we believe RL remains promising. While reward models may not provide fine-grained evaluations, they can provide useful high-level evaluations (e.g., style, tone, importance, or handling of personal or sensitive information). We therefore expect that combining SFT and RL, leveraging the strengths of both could further enhance the memory policy.
>
> > **Q2: Due to the method of constructing of entities in the memory dictionary, it seems speaker faces must be visible.**
>
> A2: The speaker's face does not need to be visible. Below are more details about how M3-Agent handles cases where the speaker's face is not seen:
>
> - The M3-Agent identifies speaker by \<voice_id\>, e.g., it generates episodic memory such as "\<voice_0\> said 'I love apples'", or semantic memory such as "\<voice_0\> likes apples".
> - These memory items are then stored in long-term memory and linked to the corresponding \<voice_id\>. Even if a face is not visible in the current clip, the M3-Agent may encounter the same voice in past or future clips where the speaker's face is visible. In such cases, it can form an association like "Equivalence: \<voice_0\>, \<face_1\>", allowing the global long-term memory to link both the face and the voice.
> - If the M3-Agent never sees the face associated with a \<voice_id\>, then the memory will only contain the \<voice_id\> and its associated information, without any facial identity.
>
> Thanks for your question. We have revised our paper (see Appendix D) to make the process description clearer.
>
> > **Q3: Is the memory agent able to merge or edit memories?**
>
> A3: We design the weighting and voting mechanism to support memory merging and editing. Specifically, when a memory item generated during the memorization process already exists in long-term memory, the corresponding node or edge is reactivated and its weight is increased (merging).
>
> When conflicting information arises, M3-Agent applies a weighted voting mechanism: nodes or edges that are activated more frequently accumulate higher weights and therefore override less frequently activated, conflicting entries. For example, suppose a clip contains the voice \<voice_0\> and the corresponding face \<face_1\> does not appear, while another face \<face_2\> appears. The model may initially incorrectly associate \<voice_0\> with \<face_2\>. However, as M3-Agent encounters more clips where \<voice_0\> and \<face_1\> cooccur, the weight of association (\<voice_0\>, \<face_1\>) will surpass that of (\<voice_0\>, \<face_2\>). Consequently, the agent corrects its memory of the face of \<voice_0\>, reflecting a process of memory editing.
>
> > **Q4: Is the agent able to call tools on its own, or is all tool use scripted beforehand?**
>
> A4: The memorization process uses tools including face detection, ASR, and speaker diarization. Because their outputs are essential for understanding most clips, these tools are invoked by default.
>
> The control process uses tools such as search_node and search_clip (see details in Table 3). After RL training, the agent can autonomously determine which search tool to use based on its current state or submitting the final answer.

---

> ### Author Response · Authors · 2025-11-19
>
> > **Q5: Ethics Concern:**
> >
> > **I don't anticipate issues; just noting the submission involves these elements and that I don't have the background to review the practices used for data collection:**
> > - **scraping youtube**
> > - **maintaining privacy of the people in the youtube videos**
> > - **human subject doing video collection; these or other people may appear in videos**
>
> A5: Thanks for your feedback.
> - For the M3-Bench-robot, all actors were fully informed about the purpose of the recordings and signed explicit consent agreements outlining data usage.
> - For M3-Bench-web, we collect videos from YouTube following existing practice [1,2,3].
>
> [1] MMBench-Video: A long-form multi-shot benchmark for holistic video understanding. NeurIPS 2024
>
> [2] OVO-Bench: How far is your video-llms from real-world online video understanding? CVPR 2025
>
> [3] Video-MME: The first-ever comprehensive evaluation benchmark of multi-modal llms in video analysis. CVPR 2025
>
> **Thank you again for your valuable comments and feedback. Please let us know if our responses address your concerns. If you have any further questions, we would be happy to address them in the discussion period.**

---

> > ### Comment · Reviewer_2Awb · 2025-11-27
> > **Thanks for the clarificatoins**
> >
> > Thanks to the authors for the clarifications (and additional experiments in the replies to other authors).
> >
> > Fundamentally I feel that this paper is systems paper, bringing together several ideas and providing a benchmark. There is nothing wrong with this! I love systems paper. These types of papers live and die by the ablations and evaluations.
> >
> > I feel the ablations here are pretty good. M3-agent may function mainly a baseline, that other methods can build on and beat.  Like reviewer urmt, I agree that M3-bench seems like the main contribution and quite useful for evaluating VLMs; especially for both robotics and web/AR usage.
> >
> > I would be curious if Reviewer urmt sees different facts that I may have missed, or if we may be looking at the same facts and simply valuing them differently (e.g. the relative value of the benchmark vs. agent). Happy to discuss further
> >
> > For now I'll keep my rating as is

---

> > > ### Author Response · Authors · 2025-11-27
> > >
> > > Thank you again for your efforts on reviewing our work and helping us polish the work. We sincerely appreciate your recognition of our contributions.

---

### Author Response · Authors · 2025-11-19
**Summary of Revisions to the Manuscript**

We sincerely thank all reviewers for their time and constructive feedback. Following their valuable comments and suggestions, we have carefully revised our paper. All revisions are highlighted in blue. The major updates in the newly uploaded version are as follows:
- Appendix D: Added detailed explanations of the long-term memory design, including how the system handles cases where only voice appears and how cross-modal associations are maintained over extended periods.
- Appendix J: Included additional cross model check for automatic evaluation. We report the main results evaluated by Gemini-1.5-Pro (**Table 15**) and Qwen3-32B (**Table 16**).
- Appendix K.1: Reported more ablation studies, including voting mehcanism, and different types of semantic memory (attributes, relations, rules) (**Table 17**).
- Appendix K.2: Reported memory item growth and QA accuracy as video length increases (**Figure 5(a)**).
- Appendix K.3: Reported memory size and QA accuracy across varying numbers of identities (**Figure 5(b)**).
- Appendix L - Hard Case in M3-Bench: Provided more details about the case study setup and summarzied the common error types along with their corresponding proportions (**Table 19**).
- Appendix L - Entity-Centric Long-Term Memory vs. Standard Memory: Provide a qualitative comparison of entity-centric memory built by M3-Agent with standard memory generated by Gemini-1.5-Pro and GPT-4o (**Table 18**).
- **Figure 1**: Update the M3-Agent architecture figure to better emphasize its contribution.
- Related Work: Supplement comparison with existing works.

---

### Meta-Review · Area_Chair_BteA · 2026-01-10

**Summary:**

The reviewers agree that this paper presents a well-engineered multimodal agent called M3-Agent. It also introduces M3-Bench for long-horizon reasoning tasks. **Reviewer 2Awb** remained positive and highlighted the practical value of the memory architecture. **Reviewers urmt and BEzR** were initially more skeptical. They raised concerns about methodological novelty and the reliance on GPT-4o for evaluation.

The rebuttal successfully addressed most of these points. The authors added cross-evaluator checks using Claude-3.5 and Qwen2.5-72B to address evaluation bias. They also clarified the memory merging and editing implementations. New streaming video experiments were added to support claims about long-term memory. Following these updates, **Reviewer BEzR** raised their score from 2 to 6. **Reviewer urmt** also acknowledged the improvements but still noted some concerns regarding novelty.

The core techniques rely on existing components, but the final integration is effective. The new benchmark and empirical results provide a useful contribution to the field.

**Reviewer Concerns:**

**Reviewer 2Awb** had a few technical questions about the memory policy and tool use. They wanted to know why the memory policy uses supervised fine-tuning while the control policy uses reinforcement learning. They also asked if the agent could edit memories or if tool use was scripted.  The authors explained that memory editing works through a weighted voting mechanism. They also confirmed that the agent makes autonomous tool calls. These clarifications satisfied the reviewer. Their positive assessment of the system design remains unchanged.

**Reviewer urmt** focused on evaluation bias and the lack of algorithmic novelty. They worried that using GPT-4o for both data generation and evaluation would skew the results. They also felt the agent relied too much on prompt engineering.  The rebuttal provided new experiments with Claude-3.5 and Qwen2.5-72B. These models showed consistent rankings and helped reduce concerns about bias. The authors also added streaming experiments to demonstrate scalability. The reviewer agreed that most issues were resolved, though they still view the novelty as limited.

**Reviewer BEzR** initially doubted the soundness of the streaming and online claims. They noted similarities to prior work like VideoAgent and requested better analysis of memory corruption. They also asked for measurements of latency and throughput.

In response, the authors provided details on the entity-centric memory design. They included new streaming experiments that showed stable computation over long sequences. These additions convinced the reviewer to change their score. They now recognize the practical value of the system despite the modest novelty.

**Reviewer Scores:**

**Reviewer 2Awb (Original: 8 → Final: 8).**
This reviewer found the benchmark and architecture to be high-quality contributions. The rebuttal answered their questions about memory merging and autonomous tools. Their score remains a strong accept.

**Reviewer urmt (Original: 4 → Final: 6).**
The cross-evaluator experiments and streaming results addressed this reviewer's main technical objections. They still have reservations about the level of novelty. However, they now lean toward a weak accept.

**Reviewer BEzR (Original: 2 → Final: 6).**
This reviewer moved from a reject to a weak accept. The new data on memory scaling and the improved evaluation cleared their strongest objections. They now value the empirical results and system integration.

---

### Decision · Program_Chairs · 2026-01-26

Accept (Poster)